# A single regulator NrtR controls bacterial NAD$^+$ homeostasis via its acetylation

Rongsui Gao[1], Wenhui Wei[1], Bachar H Hassan[2], Jun Li[3], Jiaoyu Deng[4], Youjun Feng[1,5]*

[1]Department of Pathogen Biology & Microbiology, and Department General Intensive Care Unit of the Second Affiliated Hospital, Zhejiang University School of Medicine, Hangzhou, China; [2]Stony Brook University, Stony Brook, United States; [3]Key Laboratory of Bioorganic Synthesis of Zhejiang Province, College of Biotechnology and Bioengineering, Zhejiang University of Technology, Hangzhou, China; [4]Key Laboratory of Agricultural and Environmental Microbiology, Wuhan Institute of Virology, Chinese Academy of Sciences, Wuhan, China; [5]College of Animal Sciences, Zhejiang University, Hangzhou, China

**Abstract** Nicotinamide adenine dinucleotide (NAD$^+$) is an indispensable cofactor in all domains of life, and its homeostasis must be regulated tightly. Here we report that a Nudix-related transcriptional factor, designated MsNrtR (MSMEG_3198), controls the *de novo* pathway of NAD$^+$ biosynthesis in *M. smegmatis*, a non-tuberculosis *Mycobacterium*. The integrated evidence *in vitro* and *in vivo* confirms that MsNrtR is an auto-repressor, which negatively controls the *de novo* NAD$^+$ biosynthetic pathway. Binding of MsNrtR cognate DNA is finely mapped, and can be disrupted by an ADP-ribose intermediate. Unexpectedly, we discover that the acetylation of MsNrtR at Lysine 134 participates in the homeostasis of intra-cellular NAD$^+$ level in *M. smegmatis*. Furthermore, we demonstrate that NrtR acetylation proceeds via the non-enzymatic acetyl-phosphate (AcP) route rather than by the enzymatic Pat/CobB pathway. In addition, the acetylation also occurs on the paralogs of NrtR in the Gram-positive bacterium *Streptococcus* and the Gram-negative bacterium *Vibrio*, suggesting that these proteins have a common mechanism of post-translational modification in the context of NAD$^+$ homeostasis. Together, these findings provide a first paradigm for the recruitment of acetylated NrtR to regulate bacterial central NAD$^+$ metabolism.

DOI: https://doi.org/10.7554/eLife.51603.001

*For correspondence:
fengyj@zju.edu.cn

Competing interests: The authors declare that no competing interests exist.

## Introduction

Nicotinamide adenine dinucleotide (NAD$^+$) is an indispensable cofactor of energy metabolism in all domains of life. It not only acts as an electron carrier in redox reactions (*Belenky et al., 2007*; *Magni et al., 2004*), but also functions as a co-substrate for a number of non-redox enzymes (DNA ligase [*Wilkinson et al., 2001*], NAD$^+$-dependent de-acetylase CobB/Sir-2 [*Schmidt et al., 2004*] and ADP-ribose transferase [*Domenighini and Rappuoli, 1996*]). The intra-cellular level of NAD$^+$ is dependent on the *de novo* synthesis pathway and/or its salvage or recycling route (*Gazzaniga et al., 2009*). Unlike NAD$^+$ synthesis in eukaryotes, which begins with tryptophan as a primer (*Kurnasov et al., 2003*), NAD$^+$ in most prokaryotes is produced *de novo* from the amino acid aspartate (*Kurnasov et al., 2003*). Also, certain species have evolved salvage pathway to produce NAD$^+$ (*Figure 1*) by recycling its precursor metabolites ranging from nicotinic acid (Na) (*Boshoff et al., 2008*) to nicotinamide (Nam) (*Boshoff et al., 2008*) and nicotinamide riboside (RNam) (*Rodionov et al., 2008a*; *Kurnasov et al., 2002*).

**Figure 1.** Working model for the regulation of NAD homeostasis by NrtR in *Mycobacterium*. (**A**) The genetic context of *nrtR* and its signature in *Mycobacterium* compared with the NrtR-binding sequences in *Streptococcus* (**B**) and *Mycobacterium* (**C**). (**D**) NrtR acts as an auto-repressor and represses the transcription of the *nadA-nadB-nadC* operon that is responsible for the *de novo* synthesis of the NAD$^+$ cofactor in *Mycobacterium*. (**E**) NAD$^+$ homeostasis proceeds through cooperation of a salvage pathway with *de novo* synthesis in *Mycobacterium*. Designations: *nadA,* the gene

*Figure 1 continued on next page*

*Figure 1 continued*

encoding quinolinate synthase; *nadB,* gene encoding L-aspartate oxidase; *nadC*, gene encoding quinolinate phosphoribosyltransferase; PncA, nicotinamide deaminase; PncB, nicotinate phosphoribosyltransferase; NrtR, a bifunctional transcriptional factor involved in the regulation of NAD$^+$ synthesis; ADP-R, ADP-ribose; Na, nicotinic acid; Nm, nicotinamide; Rib-P, ribose-5-phosphate; Asp, aspartate; NaMN, nicotinate mononucleotide; NAD$^+$, nicotinamide adenine dinucleotide; CobB, an NAD$^+$-consuming deacetylase.

DOI: https://doi.org/10.7554/eLife.51603.002

Tight regulation of NAD$^+$ homeostasis is needed to prevent the accumulation of harmful intermediates (*Huang et al., 2009*). In fact, three types of regulatory systems have been described for NAD$^+$ biosynthesis and/or salvage. In addition to the two well-known regulatory proteins (NadR [*Gerasimova and Gelfand, 2005*; *Raffaelli et al., 1999*] and NiaR [*Rodionov et al., 2008a*]), a family of Nudix-related transcriptional regulators (NrtR) was initially proposed via bioinformatics (*Rodionov et al., 2008b*) and recently validated in *Streptococcus suis* (*Wang et al., 2019*). The paradigm NadR protein of Enterobacteriaceae is unusual in that it has three different functional domains (*Grose et al., 2005*): i) the N-terminal transcriptional repressor domain (*Grose et al., 2005*; *Penfound and Foster, 1999*); the central domain of a weak adenylyltransferase (*Raffaelli et al., 1999*; *Grose et al., 2005*), and the C-terminal domain of nicotinamide ribose kinase (*Kurnasov et al., 2002*; *Grose et al., 2005*). NadR is a NAD$^+$ liganded regulator (*Penfound and Foster, 1999*), whereas NiaR is a nicotinic acid-responsive repressor in most species of *Bacillus* and *Clostridium* (*Rodionov et al., 2008a*). Although the prototypic NrtR possesses dual functions (Nudix-like hydrolase and DNA-binding/repressor) (*Huang et al., 2009*; *Rodionov et al., 2008b*), the NrtR homolog in *S. suis* seems to be an evolutionarily remnant regulator that lacks enzymatic activity (*Wang et al., 2019*). The phylogeny of NrtR suggests that it is widely distributed across diversified species (*Huang et al., 2009*; *Rodionov et al., 2008b*; *Wang et al., 2019*), and that its regulation of central NAD$^+$ metabolism contributes to the virulence of an opportunistic pathogen, *Pseudomonas aeruginosa* (*Okon et al., 2017*).

*Mycobacterium tuberculosis* is a successful pathogen in that it exploits flexible metabolism to establish persistent infection within the host, resulting in the disease of tuberculosis (TB) (*Bi et al., 2011*; *Shiloh and Champion, 2010*). Together with an alternative salvage route, the *de novo* synthesis of NAD$^+$ balances NAD$^+$ metabolism (*Boshoff et al., 2008*; *Bi et al., 2011*) (*Figure 1*). An earlier microbial study by *Vilcheze et al. (2005)* indicated that the removal of *ndhII*, a type II NADH dehydrogenase-encoding gene, increases the intracellular NADH/NAD$^+$ ratio, which results in phenotypic resistance to both the front-line anti-TB drug isoniazid (INH) and the related drug ethionamide (ETH). Subsequently, the *de novo* and salvage pathways of NAD$^+$ have been proposed as potential targets for anti-TB drugs (*Vilchèze et al., 2010*). Lysine acetylation is an evolutionarily conserved, reversible post-translational modification in three domains of life (*Weinert et al., 2013*). In general, the acetyl moiety is provided via two distinct mechanisms: i) Pat-catalyzed acetylation (*Starai and Escalante-Semerena, 2004*) and CobB-aided deacetylation (*Starai et al., 2002*) with acetyl-CoA as the donor of the acetyl group; and ii) the non-enzymatic action of acetyl-phosphate (AcP) donated by glycolysis (*Kakuda et al., 1994*; *Klein et al., 2007*). Not surprisingly, the lysine acetylation is linked to central metabolism via acetyl-CoA synthetase (*Xu et al., 2011*) and the biosynthesis of siderophore, an intracellular iron chelator (*Vergnolle et al., 2016*) in *Mycobacterium*. A universal stress protein (USP) is acetylated with the cAMP-dependent Pat acetyltransferase (MSMEG_5458) in *M. smegmatis* (*Nambi et al., 2010*). Nevertheless, it remains largely unclear i) how the *de novo* NAD$^+$ synthesis is regulated and ii) whether or not such regulation is connected with acetylation in *Mycobacterium*. Here, we report that this is the case. We illustrate a regulatory circuit of NAD$^+$ homeostasis by NrtR in the non-tuberculosis relative, *M. smegmatis*. More importantly, we elucidate that a post-translational modification of NrtR, acetylation of K134 in the non-enzymatic AcP manner, is a pre-requisite for its regulatory role. This might represent a common mechanism that balances the central NAD$^+$ metabolism.

## Results

### Discovery of NrtR in the context of the NAD$^+$ biosynthetic pathway

Genome context analyses suggested that the genes that encode the enzymes involved in the initial three steps of NAD$^+$ synthesis (*nadA/nadB/nadC*) are organized in a conserved manner as an operon and located adjacent to a Nudix related transcriptional regulator (*nrtR*) on the chromosome of *Mycobacterium* species (*Figure 1A*). We identified a 23-bp NrtR-binding palindrome conservatively located between the *nrtR* and *nadA/B/C* operons in mycobacteria (*Figure 1C*). The sequence of the NrtR-binding motif in *Mycobacterium* species [5′-GTTTTCGA-N7-TCGAAAAC-3′] is significantly different from that in *Streptococcus* [5′-ATA-N-TTTA-N3-TAAAA-N2-ATA-3′] (*Wang et al., 2019*) (*Figure 1B*). This may reflect the fact that these two NrtR homologs are not functionally exchangeable. Therefore, we anticipate that NrtR regulates NAD$^+$ *de novo* synthesis and coordinates it with the salvage pathway to maintain NAD$^+$ homeostasis (*Figure 1E*). The most important clue is that *de novo* NAD$^+$ synthesis is very conserved in different *Mycobacterium* species, enabling the use of *Mycobacterium smegmatis* as a model that can be used to study the regulatory mechanism for the *de novo* NAD$^+$ synthesis pathway.

### Phylogeny of NrtR

A maximum likelihood phylogenetic tree was constructed using 260 Nudix protein family representatives selected from diverse bacterial species (*Figure 2A*). The proteins carrying only a Nudix domain were removed, and 260 sequences coding for at least two protein domains were kept. Among these sequences, 38 with greater than 70% amino-acid identity were identified manually through literature mining and used for further analysis (*Figure 2B*). A common feature of the NrtR homologs is the invariant presence of the N-terminal Nudix domain (PF00293 or COG1051) fused with a characteristic C-terminal domain (PB002540), which is similar to the C-terminal part of proteins from the uncharacterized COG4111 family (*Srouji et al., 2017*). The data analyzed in this study suggest an evolutionary scenario for NrtR that includes the fusion of an ADPR-preferring Nudix hydrolase to a diversified DNA-binding domain. The phylogenetic groups found in this study show that these variable DNA-binding domains could result from the duplication of this domain and its subsequent substitution with a domain originating from a prototypical Nudix like *Tlet_0901* in *Thermotoga lettinagae* (*Zhaxybayeva et al., 2009*). Moreover, some bacterial genomes encode many Nudix proteins. For example, two probable Nudix proteins were found in *Mycobacterium* (purple clade). Interestingly, these two homologs were distributed into different phylogenetic subgroups, indicating that they have distinct origins (*Figure 2A*).

### Binding of *M. smegmatis* NrtR to cognate DNA

Using the NrtR-DNA complex (PDB: 3GZ6) as a template, structural modeling allowed us to probe the interaction of MsNrtR with its cognate DNA targets (*Figure 3A*). In total, six residues in its DNA-binding domain (namely D167, T169, N170, R173, K179, and R196) were predicted to be crucial for its DNA-binding ability (*Figure 3A–B*). Prior to biochemical analyses, the recombinant form of MsNrtR was purified to homogeneity (*Figure 3—figure supplement 1A*), and validated with both a chemical cross-linking assay (*Figure 3—figure supplement 1B*) and mass spectrometry (*Figure 3—figure supplement 1C*).

In our gel shift assays, the *nrtR* probe refers to a DNA fragment that contains a putative NrtR-recognizable palindrome [5′-GTTTTCGA-N7-TCGAAAAC-3′] (*Figure 1C*). The electrophoresis mobility shift assay (EMSA) confirmed that MsNrtR binds specifically to the *nrtR* probe (*Figure 3C*), rather than to an irrelevant DNA probe (such as the promoter of *vprA*, which encodes a response regulator of *V. cholerae*, *Figure 3—figure supplement 2*). This binding appears to be protein-dose-dependent (*Figure 3C*), which is consistent with the scenario suggested by the assay of surface plasmon resonance (SPR). In addition, SPR evaluated the binding affinity of MsNrtR to the cognate DNA probe (i.e., KD, the equilibrium dissociation constant, was around 1530 nM, *Figure 3D*). Then, six point-mutant versions of MsNrtR (D167A, T169A, N170A, R173A, K179A and R196A; *Figure 3—figure supplement 3A–B*) were also subjected to EMSA-based functional assays. In contrast to the wild-type protein (*Figure 3—figure supplement 3C*), each of these MsNrtR mutants (D167A *[Figure 3—figure supplement 3D]*, T169A *[Figure 3—figure supplement 3E]*, N170A *[Figure 3—*

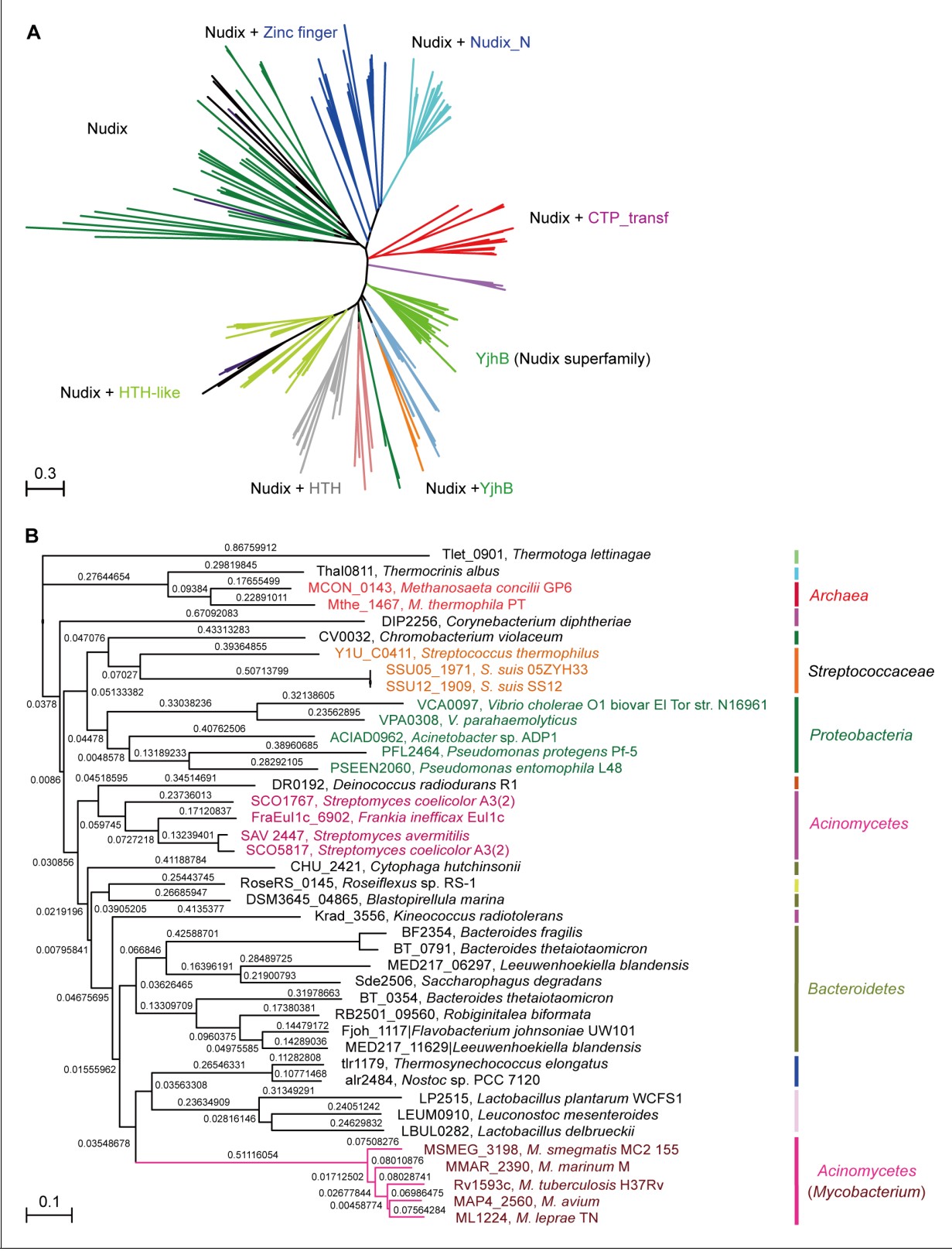

**Figure 2.** Phylogeny of NrtR proteins. (**A**) The unrooted radial phylogeny of Nudix-like proteins. A variety of distinct subclades involve homologs containing a Nudix domain alone, a Nudix domain combined with DNA-binding domains or zinc finger domains, a Nudix domain combined with a CTP-transf (Cytidylyltransferase family) domain, Nudix+Nudix_N (Nudix located at N-terminal), or a Nudix pyrophosphate hydrolase with ADP-ribose substrate preference (YjhB, Nudix+YjhB superfamily). These distinct subclades seem to coincide with known taxonomic groups with few exceptions.
*Figure 2 continued on next page*

Figure 2 continued

NrtR candidates in *Mycobacterium*, *Vibrio* and *Streptococcus* species are indicated with purple, green and orange text, respectively. (**B**) Hierarchical tree of NrtR homologs. Several distinct sub-clades are clustered in a pattern that is generally consistent with bacterial taxonomic groups. The protein-sequence-based phylogeny of NrtR homologs was inferred using the maximum likelihood method and the WAG substitution model. The evolutionary distance for each node is shown next to the branches. Gene locus tags and strain names corresponding to the protein sequences used are indicated in the figure.

DOI: https://doi.org/10.7554/eLife.51603.003

figure supplement 3F], R173A [*Figure 3—figure supplement 3G*], R196A [*Figure 3—figure supplement 3H*], and K179A [*Figure 3—figure supplement 3I*]) consistently lost their DNA-binding ability in our gel shift assays. Therefore, these six residues are indispensable for the efficient binding of MsNrtR to the cognate target gene.

## ADP-ribose disrupts interplay between NrtR and DNA

MsNrtR is an ADP-ribose pyrophosphohydrolase that belongs to the Nudix hydrolase family. Multiple sequence alignment revealed that the Nudix motif of NrtR is less conserved than is the traditional Nudix hydrolase (*Figure 3—figure supplement 4A*). Consistent with this alignment, we could not detect any apparent ADPR pyrophosphohydrolase activity in the presence of $Mg^{2+}$ or $Mn^{2+}$ (*Figure 3—figure supplement 4E*). ADP-ribose is an intermediary metabolite that is produced by glycol-hydrolytic cleavage of $NAD^+$ (*Figure 1E*), and it has been considered a highly reactive and potentially toxic molecule (*Jacobson et al., 1994*). Moreover, ADP-ribose is also a putative messenger in both eukaryotes and prokaryotes (*Huang et al., 2009*; *Rodionov et al., 2008b*; *Li et al., 1998*; *Heiner et al., 2006*). Although MsNrtR has lost its catalytic activity as an ADP-ribose pyrophosphohydrolase, we hypothesized that it retains an ability to interact with ADP-ribose. As expected, we observed that 50 mM of ADP-ribose can significantly release MsNrtR from DNA (*Figure 3—figure supplement 5*). Obviously, ADP-ribose is a ligand for the MsNrtR regulator.

## *In vivo* role of MsNrtR in $NAD^+$ synthesis

The results of PCR combined with reverse transcriptional PCR (RT-PCR) proved that the three adjacent genes *nadA*, *nadB* and *nadC* are transcribed in an operon (*Figure 4A*). Subsequently, real-time quantitative PCR (RT-qPCR) demonstrated that the removal of *nrtR* provides a 2- to 6-fold increase of the expression of the *nadA/B/C* operon compared with that in the WT (*Figure 4B*). In accordance with the qRT-PCR results (*Figure 4B*), the levels of intracellular $NAD^+$ and NADH in the Δ*nrtR* mutant are higher than those in the WT, suggesting that NrtR acts as a repressor for homeostasis of the $NAD^+$(NADH) pool (*Figure 4C–D*). To test whether or not NrtR is an auto-regulator, we fused a promoter-less LacZ to the *nrtR* promoter, giving the *nrtR-lacZ* transcriptional fusion (*Figure 4—figure supplement 1A*). Consequently, we found that deletion of *nrtR* causes a dramatic increase in the LacZ activity of *nrtR-lacZ* on agar plates (*Figure 4—figure supplement 1A*). Subsequently, we selected bacterial cultures at different growth stages (lag phase, log phase, and stationary phase, in *Figure 4—figure supplement 1B*) to compare the β-gal level of the *nrtR* promoter (*Figure 4—figure supplement 1C–E*). Intriguingly, the amplitude for auto-repression of *nrtR* is constantly around 8- to 10-fold, regardless of the growth stage of the bacteria (ranging from lag phase in *Figure 4—figure supplement 1C* and mid-log phase in *Figure 4—figure supplement 1D*, to equilibrium stage in *Figure 4—figure supplement 1E*). Thus, we concluded that the auto-repressor, NrtR, negatively regulates the expression of *nadABC* to maintain the homeostasis of the $NAD^+$(NADH) pool in *Mycobacterium*.

## Acetylation of K134 in MsNrtR

Unexpectedly, Western blotting elucidated that the recombinant MsNrtR is constantly acetylated, regardless of whether *Escherichia coli* or *M. smegmatis* was used as the expression host (*Figure 5A*). To further consolidate this observation, the MsNrtR protein was subjected to peptide mass fingerprinting through LC/MS (LTQ Orbitrap Elite) analysis. Of the five acetylated lysine sites that we identified, the Lys134 (K134) acetylation site is highly conserved in NrtR homologs from different species (*Figure 5B* and *Figure 5—figure supplement 1A*). Structural analysis suggested that K134

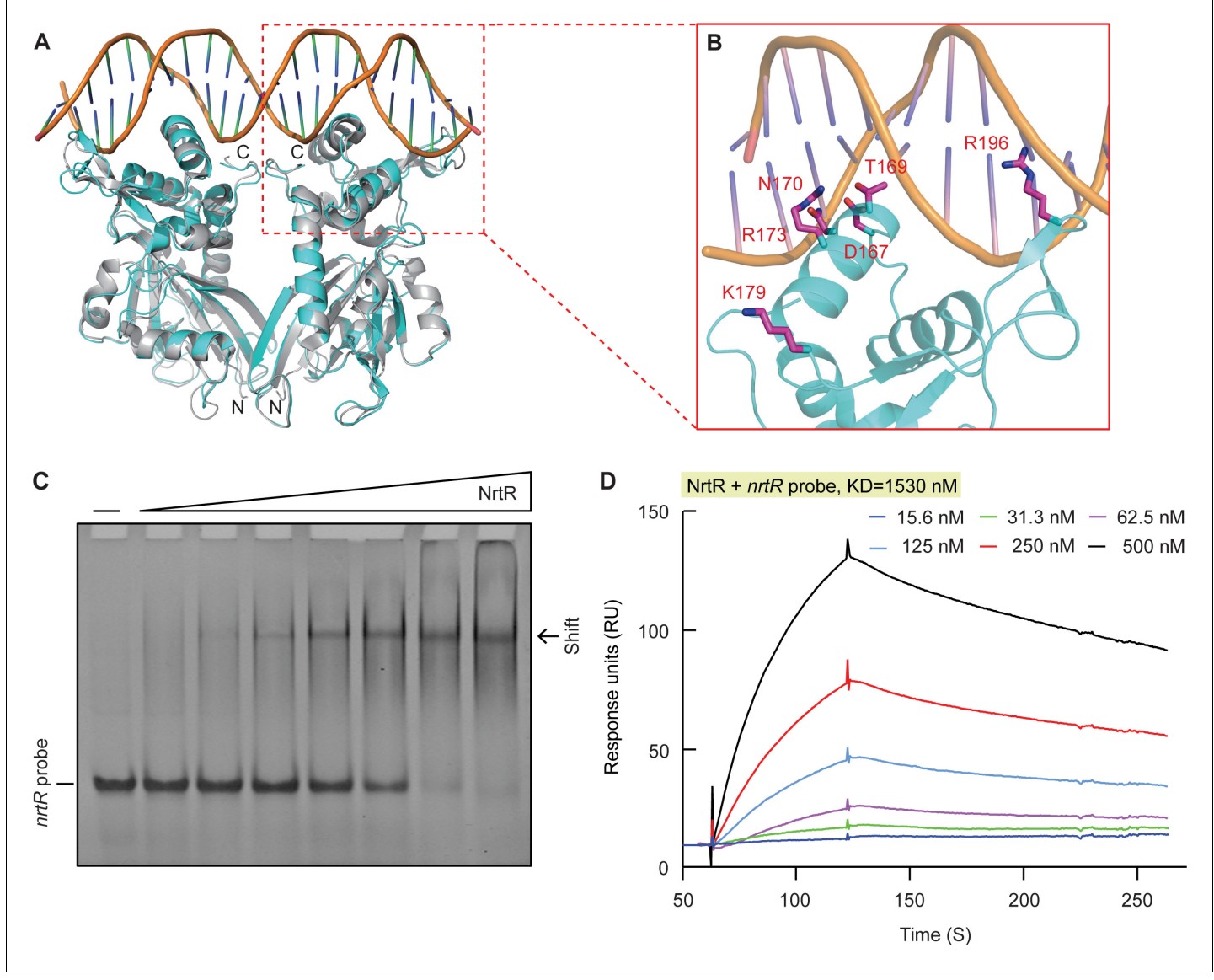

**Figure 3.** Structural and functional insights into the binding of MsNrtR to its cognate DNA target. (**A**) Structural analysis of the predicted DNA-binding motif through structural modeling of *M. smegmatis* NrtR (http://swissmodel.espasy.org/). The image shows the superposition of *M. smegmatis* NrtR with the *Shewanella oneidensis* NrtR-DNA complex (PDB: 3GZ6). MsNrtR is highlighted in cyan and soNrtR is indicated in gray. Double-stranded DNA is denoted by two orange lines. (**B**) Structural prediction of the critical DNA-binding residues of the *M. smegmatis* NrtR. The six residues (D167, T169, N170, R173, K179, and R196) that are implicated in direct or indirect contact with cognate DNA are labeled in red. (**C**) Electrophoretic mobility shift assay (EMSA)-based visualization of the interaction of MsNrtR with the *nrtR* probe. The amount of NrtR protein incubated with the DNA probe is in each lane is (left to right): 0, 0.5, 1, 2, 5, 10, 20, and 40 pmol. (**D**) Surface plasmon resonance (SPR) measurements of *M. smegmatis* NrtR binding to the *nrtR* promoter. NrtR protein at various concentrations (typically 15.625–500 nM) were injected over the immobilized DNA probe comprising of the NrtR palindrome of *nrtR* gene. KD, kd/ka, ka, association constant; kd, dissociation constant; RU, response units.

DOI: https://doi.org/10.7554/eLife.51603.004

The following figure supplements are available for figure 3:

**Figure supplement 1.** Characterization of the *M. smegmatis* NrtR.
DOI: https://doi.org/10.7554/eLife.51603.005

**Figure supplement 2.** The *M. smegmatis* NrtR cannot bind to an unrelated DNA, the *vprA* probe.
DOI: https://doi.org/10.7554/eLife.51603.006

**Figure supplement 3.** Mapping of NrtR-DNA interactions.
DOI: https://doi.org/10.7554/eLife.51603.007

**Figure supplement 4.** Identification of the ADPR pyrophosphatase activity of the *M. smegmatis* NrtR and its mutant (Q54E, K58E and D60G).
DOI: https://doi.org/10.7554/eLife.51603.008

*Figure 3 continued on next page*

*Figure 3 continued*

**Figure supplement 5.** The NAD metabolite ADP-ribose interferes with the binding of MsNrtR to cognate DNA. The image shows the electrophoretic mobility of the *nrtR* (0.2 μM) probe incubated alone (lane 1) or with purified MsNrtR (5 μM) in the absence (lane 2) and in the presence of 50 (lane 3) and 75 (lane 4) mM of ADP-ribose. The volume of the EMSA reaction system is 20 μl.

DOI: https://doi.org/10.7554/eLife.51603.009

is located at the junction between the N-terminal Nudix domain and the C-terminal HTH domain (*Figure 5B* and *Figure 5—figure supplement 1A*). This is distinct from the other six residues that have direct contact with cognate DNA (*Figure 3B* and *Figure 3—figure supplement 3B*). In addition, it seems likely that acetylation of K134 is a prevalent form because the K134A mutant protein cannot be acetylated efficiently (*Figure 5C*). Moreover, Western blot assays informed us that acetylation is present in the NrtR paralogs of *Vibrio* and *Streptococcus* (other than *Mycobacterium*, *Figure 5—figure supplement 2*).

As an important post-translational modification of protein, acetylation can adjust protein activity (e.g., DNA-binding) by affecting protein charge (and/or its conformation) in *E. coli* (*Castaño-Cerezo et al., 2014*; *Li et al., 2010*) and *Salmonella* (*Sang et al., 2016*; *Sang et al., 2017*; *Ren et al., 2016*). This prompted us to investigate the physiological role of K134 acetylation in MsNrtR. To mimic the non-acetylated form, K134 of MsNrtR was designed to mutate into arginine (R), glutamine (Q), or alanine (A), giving three mutants of MsNrtR protein (K134A, K134Q, and K134R, *Figure 5—figure supplement 1B*). Similar to the wild-type of MsNrtR, all of the mutant proteins can be purified to homogeneity and eluted at the position of dimer in our gel filtration (*Figure 5—figure supplement 1B*). This ruled out the possibility that the acetylation of K134 associated with the dimeric configuration of MsNrtR. However, the EMSA experiments verified that the DNA-binding abilities of the aforementioned three protein mutants (K134R [*Figure 5—figure supplement 1D*], K134Q [*Figure 5—figure supplement 1E*] and K134A [*Figure 5—figure supplement 1F*]) are impaired to varying degrees when compared with their parental version (*Figure 5—figure supplement 1C–G*). This hints at a possibility that an acetylation of K134 might have a physiological role in NAD$^+$ synthesis.

## Dependence of non-enzymatic AcP in MsNrtR acetylation

In general, the reversible acetylation of MsNrtR falls into one of two categories: enzymatic action by the acetyltransferase Pat and a non-enzymatic acetyl phosphate (AcP)-dependent m (*Figure 6A*) (*Ren et al., 2016*; *Ren et al., 2019*). Of note, both of these processes can be reversed by the deacetylase CobB (*Figure 6A*) (*Ren et al., 2016*; *Ren et al., 2019*). To address possible origin of K134 acetylation, we integrated an *in vitro* chemical assay and a genetic exploration *in vivo* (*Figure 6B–I*). To test the relevance of K134 acetylation to the enzymatic action of Pat/CobB, we deleted the *pat* (MSMEG_5458)/*cobB* (MSMEG_5175) gene (*Figure 6A*) from *M. smegmatis* by homologous recombination. As expected, the USP (universal stress protein, MSMEG_4207) is validated as the positive control that requires Pat for its enzymatic acetylation (*Figure 6B*). By contrast, we detected no difference in the acetylation levels of the MsNrtR proteins of WT, Δ*pat* and Δ*cobB* (*Figure 6C*). This might rule out the possibility of Pat-catalyzed acetylation of MsNrtR. Then, we wondered whether or not K134 acetylation of MsNrtR proceeds via AcP-dependent route (*Figure 6D–I*). First, we established an *in vitro* system in which MsNrtR was incubated with AcP (*Figure 6D–E*), an intermediate product of glycolysis (*Figure 6J*). This showed clearly that the percentage of acetylated NrtR increased in a time-dependent manner (*Figure 6D and F*). This acetylation also occurred in an AcP-dose-dependent manner (*Figure 6E and G*). Obviously, these findings constitute *in vitro* evidence that AcP donates the acetyl group for the post-translational acetylation of NrtR (*Figure 6D–G*). Second, we secured *in vivo* evidence by knocking-out the AcP-pathway-encoding genes *ackA* and *pta* (*Figure 6A and J*). The acetylation level of MsNrtR in the double mutant of *M. smegmatis* (Δ*ackA* +Δ*pta*), is reduced 4–5-fold when compared with its parental strain (*Figure 6H–I*). This finding is similar to those of *Weinert et al. (2013)* working on NrtR acetylation in *E. coli* under different growth conditions (induced with glucose or acetate). In *M. smegmatis*, the removal of a single *ackA* only slightly repressed the acetylation of MsNrtR growing under inducing conditions of either 0.2% glucose or 1.0% acetate (*Figure 6H–I*). Along with the *in vitro* data, the *in vivo* evidence allowed us to

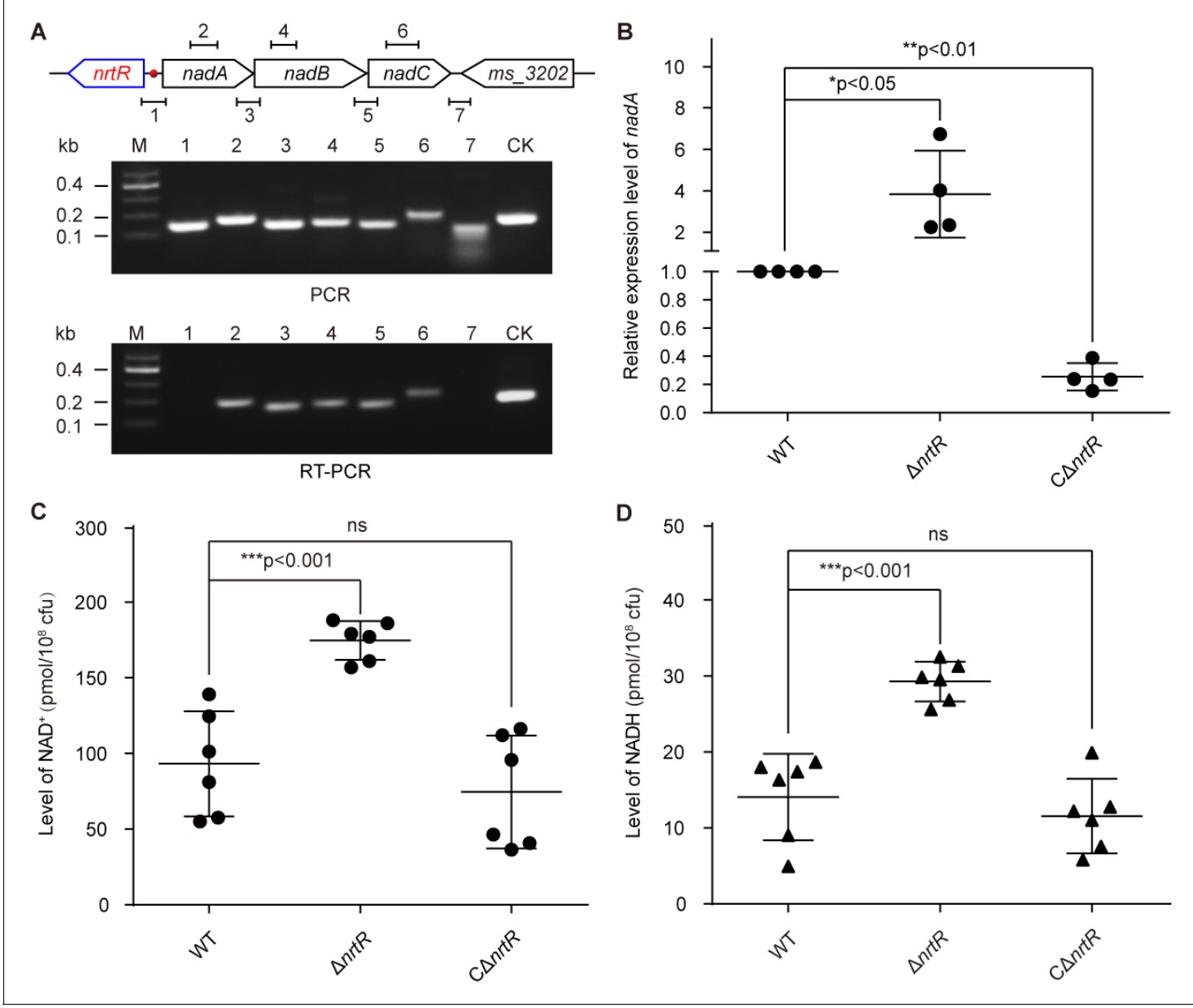

**Figure 4.** NrtR is a repressor for the *nadABC* operon that is responsible for NAD+ and NADH concentration in *M. smegmatis*. (**A**) Genetic organization and transcriptional analyses of the *nrtR* and its neighboring *de novo* NAD+ synthesis genes. The arrows represent open reading frames, and the numbered short lines (1 to 7) represent the specific PCR amplicons that were observed in the following PCR and RT-PCR assays (in the bottom panels). PCR and RT-PCR were applied to analyze the transcription of the putative NAD+ *de novo* synthesis loci. The primer numbering was identical to that shown in the top panel. CK (control) denotes the 16S rDNA. (**B**) RT-qPCR analyses of *nad* operon expression in the wild-type strain and in the Δ*nrtR* mutant and *nrtR* complementary strains. RT-qPCR experiments were performed at least three times and the data were expressed as means ± standard deviations (SD). The p-value was calculated using one-way ANOVA along with Tukey's test. *p<0.05 and **p<0.01. Comparison of the intra-cellular level of NAD+ (**C**) and NADH (**D**) among the WT, Δ*nrtR* and CΔ*nrtR* strains. Each dark circle or triangle represents an independent experiment. The data are shown as means ± SD. The statistical significance of differences among WT, Δ*nrtR* and CΔ*nrtR* was determined by Student's t test and by ANOVA with heterogeneous variances. ***p<0.001; ns, no significant difference.

DOI: https://doi.org/10.7554/eLife.51603.010

The following figure supplement is available for figure 4:

**Figure supplement 1.** *In vivo* evidence that MsNrtR is an auto-repressor.
DOI: https://doi.org/10.7554/eLife.51603.011

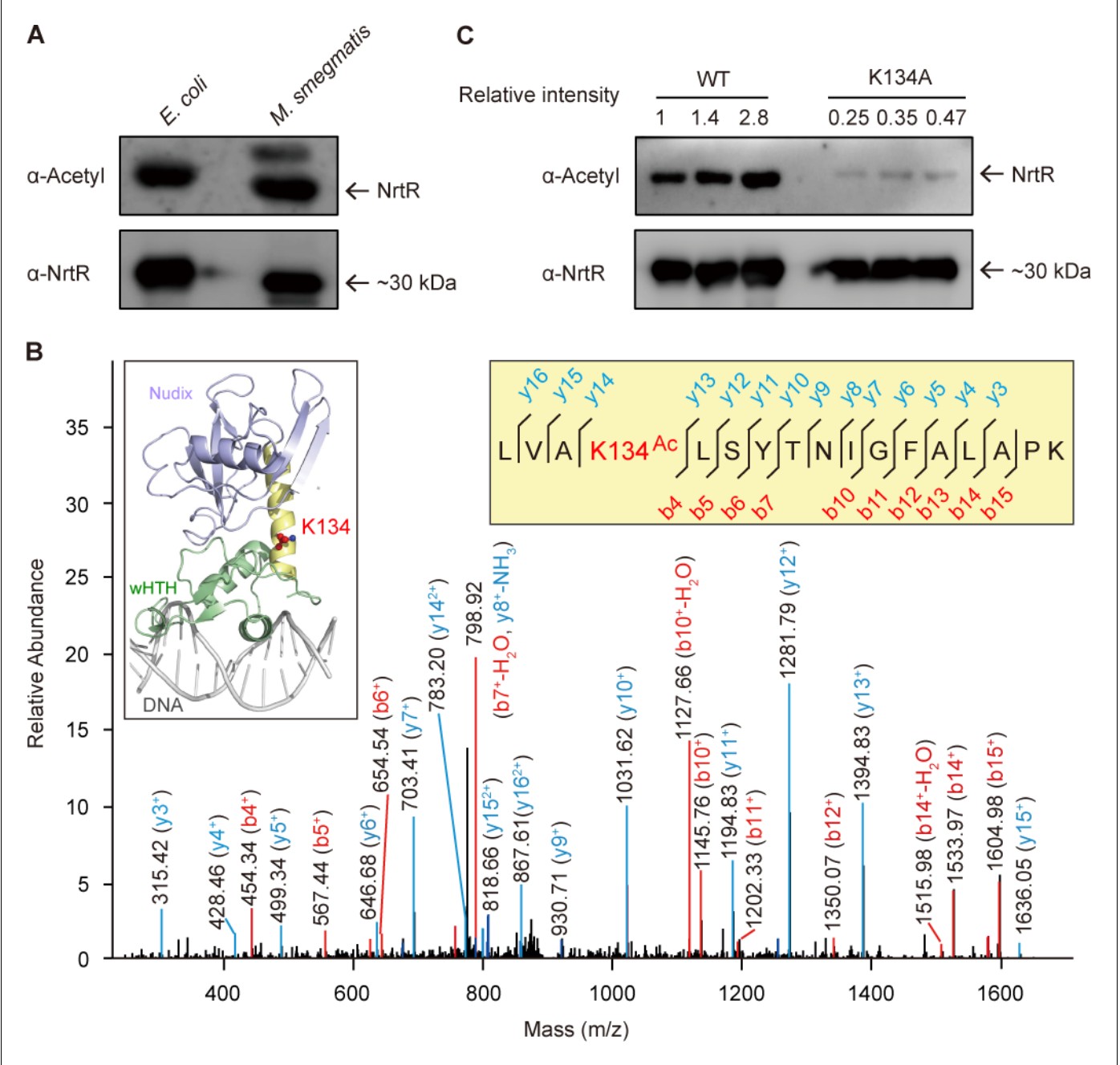

**Figure 5.** The discovery of acetylation of K134 in MsNrtR. (**A**) Use of Western blot to probe the acetylation of recombinant MsNrtR protein in both *E. coli* and *M. smegmatis*. The two forms of recombinant NrtR protein were purified from *E. coli* BL21 and *M. smegmatis*, and analyzed by western blotting using both anti-acetyl-lysine antibody (α-Acetyl) and poly anti-MsNrtR rabbit serum. The bigger version of MsNrtR is produced by the pET28 expression plasmid in *E. coli*, whose N-terminus is fused to the 6xHis-containing tag of 23 residues (***Supplementary file 1***). By contrast, the smaller version of MsNrtR is generated by pMV261 in *M. smegmatis*, which is only tagged with C-terminal 6xHis. The altered molecular mass (~2 kDa) is the reason why the migration rate of protein electrophoresis differs slightly for the two MsNrtR versions. A representative result is given from three independent trials. (**B**) The discovery of a unique Lys134 acetylation site in MsNrtR. A LC/MS spectrum reveals that a charged peptide (LVAkLSYTNIGFALAPK) of MsNrtR bears an acetylated lysine (K134$^{Ac}$). The sequence depicted in the yellow box illustrates the K134 site of acetylation in the context of the modeled structure of MsNrtR-DNA. (**C**) The mutation of K134A results in reduced acetylation of MsNrtR in *M. smegmatis* MC$^2$ 155 (***Magni et al., 2004***). A representative result from three independent experiments is given.

DOI: https://doi.org/10.7554/eLife.51603.012

The following figure supplements are available for figure 5:

**Figure supplement 1.** Dependence on K134 acetylation in the binding of MsNrtR to a cognate DNA target.

*Figure 5 continued on next page*

*Figure 5 continued*

DOI: https://doi.org/10.7554/eLife.51603.013
**Figure supplement 2.** Acetylation is ubiquitous in three bacterial NrtR proteins.
DOI: https://doi.org/10.7554/eLife.51603.014
**Figure supplement 3.** Construction and identification of NrtR K134 point mutants on the *M. smegmatis* chromosome.
DOI: https://doi.org/10.7554/eLife.51603.015

conclude that the K134 acetylation of MsNrtR is physiologically dependent on the AckA/Pta-containing route for AcP formation (*Figure 6J*).

## Physiological roles of K134 acetylation

To further investigate the *in vivo* role of K134 acetylation in the regulatory function of NrtR (*Figure 7*), we engineered *M. smegmatis* mutants carrying point-mutations of K134 (namely K134A, K134Q, and K134R) on chromosomal *nrtR* (*Figure 5—figure supplement 3A–C*). All the point-mutants of K134 were confirmed with direct DNA sequencing (*Figure 5—figure supplement 3C*). Also, Western blot was applied to prove that the mutated proteins are well expressed *in vivo* (*Figure 5—figure supplement 3D*). As expected, the removal of *nrtR* (positive control) gave around a ten-fold increase in the β-gal level of *nrtR-lacZ* transcriptional fusion (*Figure 7A*). In general agreement with the positive control, the K134Q mutation led to a three-fold upregulation in *nrtR* transcription (*Figure 7A*), Similar scenarios were also observed for the other two point-mutants, K134A and K134R, despite the lower level of (close to two-fold) of regulatory dysfunction in the control of *nrtR* transcription (*Figure 7A*). RT-qPCR assays showed that functional impairment of the K134 acetylation site can increase the transcriptional level of the *nad*ABC operon 2–3-fold (*Figure 7B*). More importantly, the pool of intra-cellular $NAD^+$ in the K134 point-mutants accumulated to a level that was 2–3-fold that in the wild-type strain (*Figure 7C*). A similar observation occurs with NADH (*Figure 7D*).

Given that i) the K134 acetylation of NrtR involves the control of the cytosolic $NAD^+$ pool (*Figure 7*) and ii) the acetylation of NrtR is AcP-dependent (*Figure 6*), we hypothesized that the AckA/Pta-including AcP pathway contributes to the NrtR-mediated regulation of the cytosolic $NAD^+$ pool (*Figure 8A*). As anticipated, inactivation of the AcP pathway (especially in the Δ*pta*+Δ*ackA* double mutan having AcP level of ~10 μM, largely lower than that of the parental strain, ~460 μM) led to a significant increase in the β-gal level of *nrtR-lacZ* transcriptional fusion (*Figure 8B*). In particular, the cytosolic pools of both $NAD^+$ (*Figure 8C*) and NADH (*Figure 8D*) were increased 2.5–3-fold in the double mutant (Δ*pta*+Δ*ackA*), which is deficient in the AcP pathway. Therefore, the data suggest that the AcP-dependent acetylation of K134 is necessary for NrtR to regulate the homeostasis of $NAD^+$ in *Mycobacterium*.

## Discussion

It has been estimated that 17% of the enzymes of central metabolism that are essential for the survival of *M. tuberculosis* require the $NAD^+$ cofactor (*Beste et al., 2007*), regardless of the organism's state of latency or active-replication (*Rodionova et al., 2014*). It is rational that the $NAD^+$ metabolic pathway and its regulatory mechanism have been recognized as an attractive target for the development of new anti-TB therapeutics (*Rodionova et al., 2014*). Although NrtR, as the third regulator of $NAD^+$ metabolism (*Rodionov et al., 2008b*), has been described *in vitro* in different species such as *Shewanella* (*Huang et al., 2009*), the data we show here provide the first relatively full picture of the regulatory circuits of $NAD^+$ synthesis involving NrtR, an evolutionarily distinct regulator, in a non-pathogenic *M. smegmatis* (*Figure 1*). Although it structurally comprises an N-terminal Nudix domain and a C-terminal Helix-Turn-Helix motif (*Huang et al., 2009*), MsNrtR retains only the ability to bind cognate DNA (*Figure 3*) and loses its ADP-ribose hydrolase activity (*Figure 3—figure supplement 3*). There are mutations at three residues (GX$_5$EX$_7$R$\underline{Q}$UX$\underline{E}$$\underline{K}$X$\underline{D}$U) (*Carreras-Puigvert et al., 2017*) in the Nudix motif of MsNrtR that we attempted to reverse, but we were unable to recover the enzymatic function of the protein (*Figure 3—figure supplement 4*). It seems likely that natural selection/ evolution has rendered the hydrolyzing ADP-ribose NrtR inactive in order to avoid functional

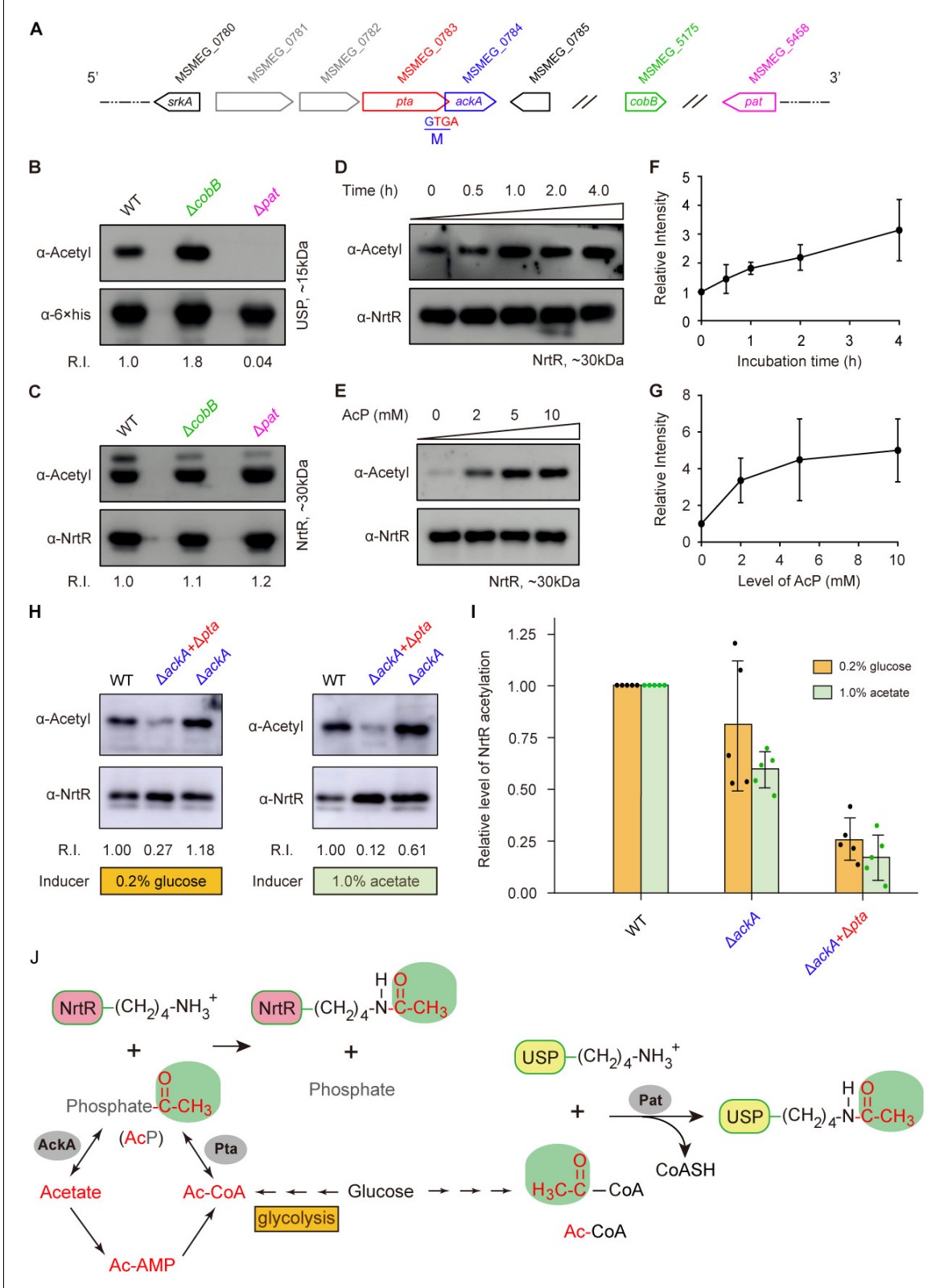

**Figure 6.** Acetyl phosphate-mediated acetylation of MsNrtR. (**A**) Genetic context of the two types of acetylation pathways in *M. smegmatis*. The two genes *pat* (MSMEG_5458) and *cobB* (MSMEG_5175) are responsible for the reversible enzymatic route of acetylation. The two loci *ackA* (MSMEG_0784) and *pta* (MSMEG_0783) participate in the non-enzymatic AcP pathway. Of note, *ackA* and *pta* are two overlapping loci that appear as an operon. (**B**) The acetylation levels of USP are dependent on the Pat/CobB-requiring enzymatic route in *M. smegmatis*. USP denotes the universal stress protein

*Figure 6 continued on next page*

*Figure 6 continued*

(MSMEG_4207). Using the recombinant plasmid pMV261-*usp*, the 6 × His tagged USP protein was expressed in wild-type *M. smegmatis* and its derivatives (Δ*cobB* and Δ*pat*). As a result, the acetylation levels of the purified USP proteins were detected with the pan anti-acetyl lysine antibody (α-Acetyl) and an anti-6 ×his antibody was used as a loading control. A representative result for three independent experiments displayed. (C) The acetylation levels of NrtR are not distinguishable in the three strains of *M. smegmatis* (wild-type, Δ*cobB* and Δ*pat*). 6 × His tagged MsNrtR was expressed in the three strains described for panel (B) using pMV261-*nrtR*. The acetylation levels of the purified MsNrtR proteins were determined using the α-Acetyl antibody, and anti-NrtR antiserum (α-NrtR) acted as a loading control. Western blots were conducted in triplicates. (D) Western-blot-based detection of the *in vitro* non-enzymatic acetylation of MsNrtR using AcP as the phosphate donor. Acetylation of MsNrtR by AcP (10 mM) was measured by incubating MsNrtR and AcP for 0, 0.5, 1, 2 and 4 hr at 37°C. The concentration of NrtR was determined by Western blot with anti-NrtR serum as a primary antibody (lower panel). (E) Acetylation of MsNrtR is AcP dose-dependent. MsNrtR was incubated with different levels of AcP (0, 2, 5 and 10 mM) for 2 hr at 37°C. (F) Altered acetylation of MsNrtR as incubation progresses over time with constant AcP. Acetylation was quantified using Image J software and normalized to the signal at 0 hr. (G) MsNrtR acetylation at various levels of AcP. Data were measured with Image J software and normalized to the signal at 0 mM AcP. Data are shown as mean ± standard deviation (SD). (H) *In vivo* evidence that the AcP pathway is associated with NrtR acetylation. In addition to the parental strain, a single mutant (Δ*ackA*) and double mutant (Δ*ackA*+Δ*pta*) were used to prepare the recombinant MsNrtR proteins with varied levels of acetylation. Of note, the bacterial growth conditions were supplemented with an inducer of 0.2% glucose or 1.0% acetate recommended by *Weinert et al. (2013)*. The Western blot was performed as described for panels (B) and (C). Representative results of three or more independent experiments are shown. (I) Contribution of the AckA and Pta-requiring AcP pathway to NrtR acetylation. The acetylation signal was quantified using Image J software, and the density in the WT was normalized as 1. Each dot denotes a Western blot experiment. (J) Working model for non-enzymatic acetylation of MsNrtR in a metabolic context, and the working model for the enzymatic acetylation of USP. Abbreviations: MsNrtR, *M. smegmatis* NrtR; AcP, Acetyl-phosphate; AcAMP, Acetyl-AMP; Ac-CoA, acetyl-CoA; USP (MSMEG_4207), universal stress protein (130aa); Pta (MSMEG_0783), phosphate acetyltransferase (692aa); and AckA (MSMEG_0784), acetate kinase (376aa).

DOI: https://doi.org/10.7554/eLife.51603.016

redundancy. This prediction is in part (if not entirely) explained by the genome-wide distribution of 29 predicted proteins of the Nudix hydrolase family in *M. smegmatis* (*Supplementary file 3*).

A similar scenario in which the *S. suis* NrtR has an inactive Nudix hydrolase domain (*Wang et al., 2019*) allowed us to further hypothesize that most members of the Nudix-related regulator family might initially recruit ancient Nudix hydrolase as a signaling module, and then un-necessitate its enzymatic function while retaining its regulatory role as an evolutionary relic (*Wang et al., 2019*). The diversity of genomic organization of *nrtR* and its neighboring regulatory loci highlights that this gene is being subjected to dynamic domestication (*Rodionov et al., 2008b*). Evidently, the *nrtR* is integrated into the '*nadR-pnuC-nrtR*' cluster in the human pathogen *S. suis* 2, assuring a regulated salvage/recycling pathway (*Rodionov et al., 2008b*; *Wang et al., 2019*). By contrast, the *nrtR* on the opposite strand is adjacent to an operon of *nadA/B/C* (the intergenic region of which contains a NrtR-recognizable site; *Figures 1A* and *4A*), guaranteeing the control of *de novo* NAD$^+$ synthesis (*Figure 1D–E*). Indeed, the NrtR is somewhat promiscuous because it modulates xylose (e.g., *xylBAT* of *Bacteroides*) and arabinose (e.g., *araBDA* of *Flavabacterium*) utilization in rare microorganisms (*Rodionov et al., 2008b*). Consistent with earlier descriptions (*Rodionov et al., 2008b*), the NrtR orthologs of *S. suis* (*Wang et al., 2019*) and *M. smegmatis* (*Figure 3—figure supplement 5*) proved to be antagonized by ADP-ribose in the binding to cognate DNA targets. By contrast, the NrtR (also named NdnR) of *Corynebacterium glutamicum*, a close relative of *Mycobacterium*, has surprisingly been found to exhibit more affinity to binding the DNA probe in the presence of NAD$^+$ (*Teramoto et al., 2012*). This is probably explained by the varied configuration of signaling module within different NrtR orthologs. Because the NrtR of *Pseudomonas* participates in the fitness and virulence of this pathogen within mice (*Okon et al., 2017*), it is very interesting to wonder how NrtR and its regulated route of NAD$^+$ synthesis contribute to the survival and chronic infection of *Mycobacterium* within the host environment.

Protein acetylation is a ubiquitous form of post-translational modification in prokaryotes (*Ren et al., 2017*), which is implicated in central metabolism and even bacterial pathogenicity (*Ren et al., 2019*; *Sang et al., 2016*). A global acetylome analysis of *M. tuberculosis* by Ge and cow-orkers (*Liu et al., 2014*) identified almost 137 unique acetylated proteins that are involved in diverse biological processes, some of which had undergone lysine acetylation. In this study, we are first to discover the lysine acetylation (K134) at the junction between the N-terminal Nudix domain and the C-terminal wHTH domain of MsNrtR (*Figure 5*). It is unusual, but not without any precedent. In fact, our research group very recently affirmed the presences of such a modification of K47 in the *M. smegmatis* BioQ that regulates biotin metabolism (*Tang et al., 2014*; *Wei et al., 2018*). Given that

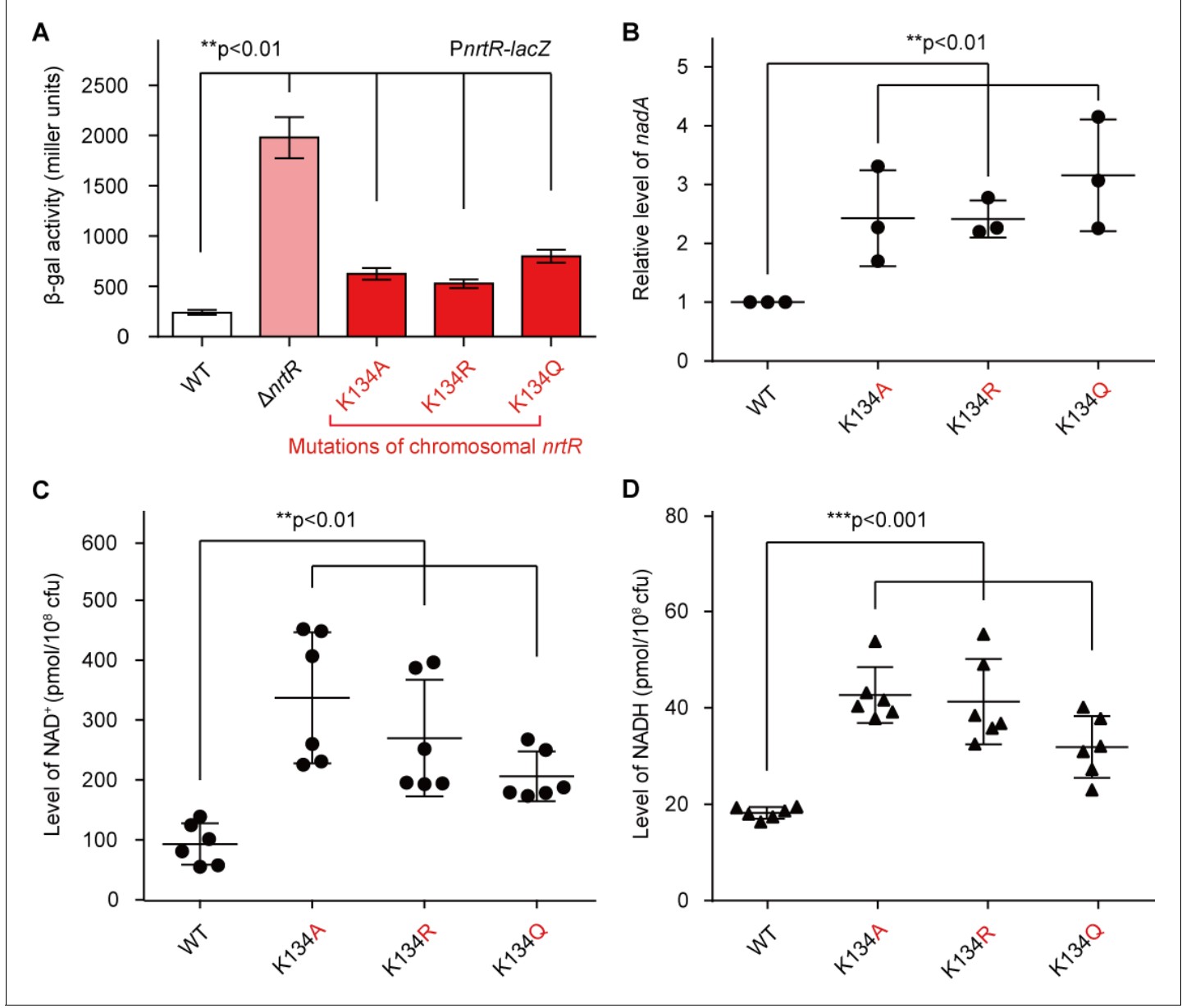

**Figure 7.** Acetylation of K134 in MsNrtR determines its role in the homeostasis of the intracellular NAD$^+$ pool. (**A**) Genetic assays for the auto-repression of *nrtR* using P*nrtR-lacZ* transcriptional fusion. These results suggest that functional impairments in K134 acetylation lead to de-repression of *nrtR*, as does the removal of *nrtR*. (**B**) RT-qPCR analyses of the transcription of the *nad* operon in the mutant carrying a point-mutation of K134 in *nrtR* (K134A, K134R and K134Q). Levels of intracellular NAD$^+$ (**C**) and NADH (**D**) in the WT *nrtR* strain and its point-mutants (K134A, K134R and K134Q). All of the experiments were performed at least three times, and the data are presented as means ± SD. The p-values were calculated using one-way ANOVA along with Tukey's test.

DOI: https://doi.org/10.7554/eLife.51603.017

The following figure supplement is available for figure 7:

**Figure supplement 1.** Determining the number of live cells at OD600 during the exponential phase.
DOI: https://doi.org/10.7554/eLife.51603.018

K134 is conserved in NrtR orthologs of different origins (**Figure 7—figure supplement 1A**), we hypothesized that it is a common hallmark for NrtR. However, this requires further experimental demonstration. Evidently, it is likely that a single lysine acetylation maintains the pools of two distinct cofactors (biotin and NAD$^+$) by modifying a certain regulator. In light of the facts that i) mycobacterial NAD$^+$ metabolism is regarded as an promising drug target (*Bi et al., 2011*; *Rodionova et al.,*

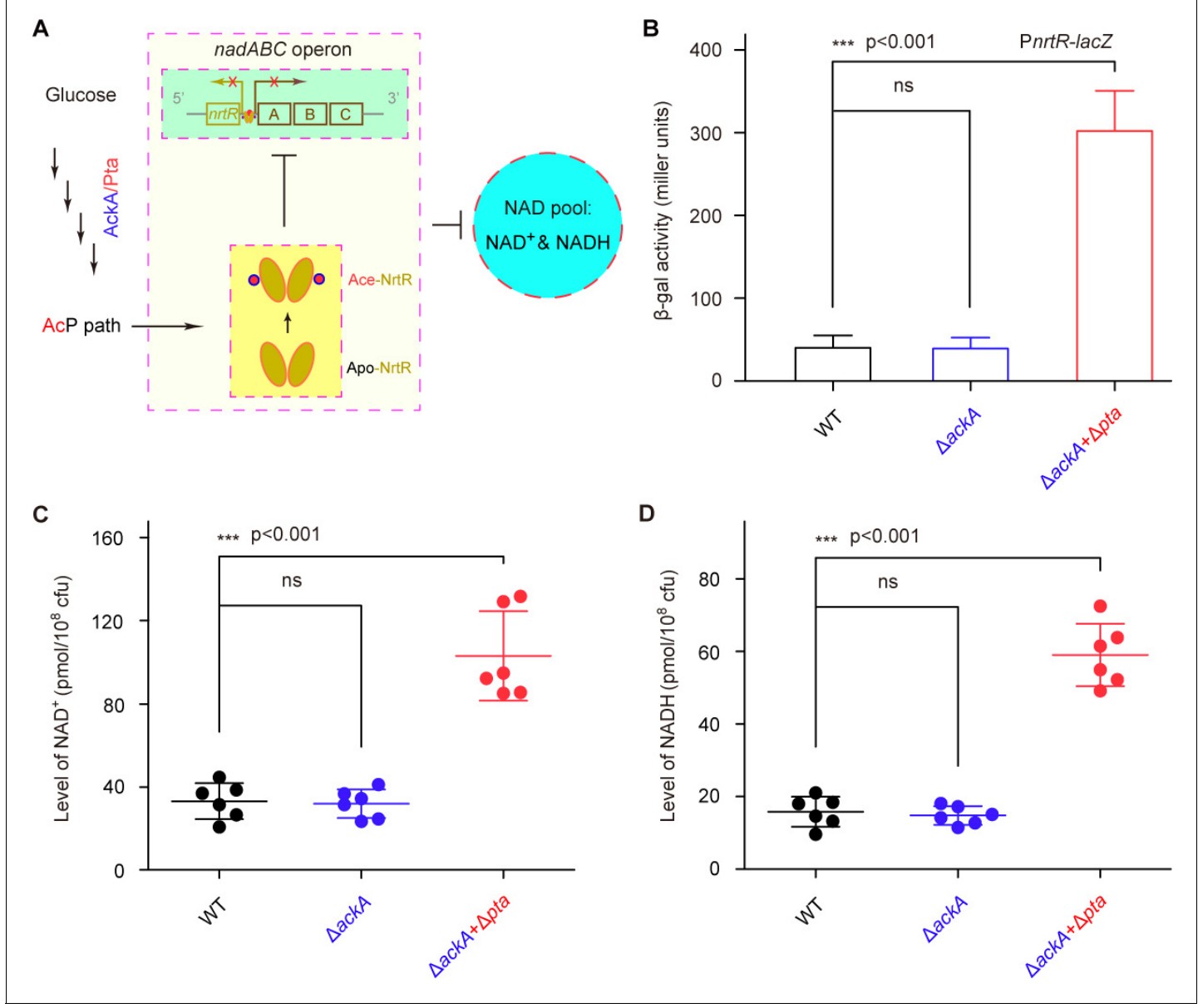

**Figure 8.** AcP-pathway-dependent repression of NAD$^+$ synthesis by NrtR in *M. smegmatis*. (A) Scheme for the maintenance of NAD$^+$ homeostasis by AcP-dependent NrtR acetylation (B) The removal of *ackA* and *pta* from *M. smegmatis* increases the β-gal level of P*nrt-lacZ* transcriptional fusion in the presence of 0.2% glucose in the growth medium. (C) The level of the cytosolic NAD$^+$ pool is elevated in the double mutant of *M. smegmatis* (Δ*ackA* + Δ*pta*) in the growth condition with 0.2% glucose added. (D) The inactivation of the AcP path gives an increase of ~3 fold in the cytosolic NADH pool Three strains of *M. smegmatis* (WT, Δ*ackA*, and Δ*ackA* + Δ*pta*) were cultivated in 7H10 medium supplemented with 0.2% glucose. No less than three independent measures were carried out, and the values presented here are averages ± SD. *, p<0.001; ns, no significance.
DOI: https://doi.org/10.7554/eLife.51603.019

2014) and ii) the synthesis and utilization of biotin is necessary for survival and infectivity of intracellular pathogens (*Park et al., 2015*; *Bockman et al., 2015*; *Woong Park et al., 2011*), we hypothesized that lysine acetylation plays an indispensable role in bacterial virulence. This is generally consistent with the K201 acetylation of PhoP, a response regulator of the two-component system in *Salmonella* virulence (*Ren et al., 2016*; *Ren et al., 2019*). Given that the acetylation of NrtR paralogs is also detected in two additional pathogenic species, the Gram-negative *V. cholerae* and Gram-positive *S. suis* (*Figure 5—figure supplement 2*), it is possible that the lysine acetylation of NrtR

represents an evolutionarily conserved mechanism by which a group of pathogens develop successful infections within the nutrition-limited tough host niche.

In summary, the functional definition of K134 acetylation in MsNrtR updates our understanding of the homeostasis of NAD$^+$, an indispensable coenzyme. This evidence provides an alternative paradigm for the development of anti-TB virulence lead drugs that can impair crosstalk between the nutritional/restricted virulence factor (NAD$^+$) and NrtR by disrupting the acetylation-requiring regulatory system (*Figure 8A*).

## Materials and methods

### Bacterial strains, plasmids and growth conditions

The bacterial species used in this study include *E. coli* and *M. smegmatis* (*Supplementary file 1*). Strains were cultured as described previously (*Tang et al., 2014*; *Wei et al., 2018*). pET28a-*nrtR* and pMV261-*nrtR* were constructed as follows. *nrtR* was cloned into the pET-28a expression vector between BamHI and XbaI restriction sites. The resulting plasmid contained the *nrtR* gene fused to a hexahistidine-tag sequence at the N-terminus and was transformed into *E. coli* BL21(DE3) for heterologous expression of NrtR. The mycobacterial expression plasmid pMV261-*nrtR* was constructed by cloning the *nrtR* gene fused to a His-tag sequence at the C-terminus via BamHI and SalI sites. This recombinant plasmid was then electro-transformed into *M. smegmatis* MC$^2$ 155 (*Magni et al., 2004*) for endogenous production of NrtR. Overlap PCR was utilized for site-directed mutagenesis of the *nrtR* gene using specific primers (*Supplementary file 2*).

### Protein expression, purification and identification

Wild-type MsNrtR and its point mutants were overexpressed in *E. coli* or *M. smegmatis* MC$^2$ 155 (*Magni et al., 2004*). The expression of proteins in *E. coli* (BL21) was induced by the addition of 0.5 mM isopropyl β-d-1-thiogalactopyranoside (IPTG) (*Gao et al., 2017*). For protein purification, the cells were harvested by centrifugation and lysed by sonication. The clarified lysate was loaded onto a Ni-nitrilotriacetic acid (Ni-NTA) column (Qiagen) and eluted with 150 mM imidazole (*Gao et al., 2017*). The protein preparation was further purified by gel filtration through a Superdex 75 10/300 column (GL, GE Healthcare) and the protein purity was judged with 12% SDS-PAGE.

The recombinant MsNrtR expressed in *E. coli* and *M. smegmatis* was separated by SDS-PAGE gel and subjected to peptide mass fingerprinting with Liquid Chromatography (LC)-mass spectrometry (MS) (*Wei et al., 2018*; *Gao et al., 2016a*). The resultant polypeptides were separated by the EASY-nLC HPLC system (Thermo Scientific, USA) and detected using a Thermo Fisher LTQ orbitrap elite mass spectrometer (Thermo Scientific, USA). The MS spectrum containing the possible site of acetylation was detected and assigned by Mascot 2.2 (*Ren et al., 2016*). To further visualize the solution structure, 6 × His tagged MsNrtR protein was subjected to chemical cross-linking assays with the cross-linker of ethylene glycol bis-succinimidylsuccinate (Pierce) as we earlier described (*Feng and Cronan, 2010*).

### Electrophoretic mobility shift assays

The interaction of MsNrtR with its DNA target was specified by gel shift assay (*Gao et al., 2017*). The DNA probe containing an MsNrtR-recognizable palindrome (designated *nrtR*) was generated by annealing two complementary primers (*nrtR*-probe-F and *nrtR*-probe-R) in TEN buffer (10 mM Tris-HCl, 1 mM EDTA, 100 mM NaCl [pH 8.0]) (*Gao et al., 2016b*). The DNA probe was mixed with purified NrtR protein in EMSA buffer (50 mM Tris-HCl [pH 7.5]; 10 mM MgCl$_2$; 1 mM DDT; 100 mM NaCl) and incubated at room temperature for about 30 min (*Wei et al., 2018*). The DNA–protein complexes were separated on native 8% native polyacrylamide gels, and the shifted DNA bands were visualized by staining with ethidium bromide (EB) (*Gao et al., 2017*).

### Surface plasmon resonance

To evaluate the parameters of binding between NrtR and its DNA target, surface plasmon resonance (SPR) was employed using a Biacore3000 instrument (GE Healthcare) at 25°C. A biotinylated *nrtR* probe was injected onto the flow cells of a SA sensor chip at a flow rate of 10 μl/min until the calculated amount of DNA had been bound, giving a 34 RU maximum. All of the SPR experiments were

conducted in the running buffer (50 mM Tris-HCl [pH 7.5], 150 mM NaCl and 0.005% (v/v) Tween 20) at a flow rate of 30 µl/min (*Gao et al., 2017*). A series of dilutions of protein samples were injected and passed over the chip surface for 2 min. The dissociation phase was followed for 3 min in the same buffer, and the surface was then regenerated with 0.025% SDS for 24 s. Kinetic parameters were analyzed using a global data analysis program (BIA evaluation software).

### Assays for β-gal activity

Transcriptional levels were measured using *lacZ* transcriptional fusions carried on plasmid pMV261 in *M. smegmatis* (WT and Δ*nrtR*). Cells from log-phase cultures grown in LB broth containing 0.2% glycerol, 0.05% Tween-80, and 50 µg/ml kanamycin were collected by centrifugation, washed twice with RB medium supplemented with 0.05% Tween-80. and re-suspended in Z-buffer for measurement of β-gal activity (*Miller, 1992*; *Feng and Cronan, 2009a*; *Feng and Cronan, 2009b*). Data were obtained from three independent trials and presented as a means and standard deviations (SD).

### Generation of chromosomal knock-out and knock-in strains

The knock-out and knock-in strains of the *M. smegmatis nrtR* (MSMEG_3198) gene were generated using the homologous recombination method as described before (*Tang et al., 2014*; *Yang et al., 2012*). A suicide plasmid was constructed by cloning the knock-out or knock-in fusion PCR products (*Supplementary file 1*) into pMind between NheI and PacI, followed by the insertion of a *sacB-lacZ* cassette as a selection marker at the PacI site (*Supplementary file 1*). The recombinant plasmid was electroporated into competent cells of *M. smegmatis* (wild-type or *nrtR* deletion mutant) and plated on LB medium containing 100 µg/ml X-gal and 50 mg/L kanamycin for screening of single-crossover mutant strains. Single colonies were picked and inoculated into kanamycin-free LB broth, 37°C, 220 rpm for 24 hr. The incubated culture was plated on LB medium containing 100 µg/ml X-gal and 10% sucrose. The white colonies representing allelic-exchange mutants were picked and identified by multiplex-PCR and direct DNA sequencing. The mutants of *M. smegmatis* include an in-frame deletion mutant (Δ*nrtR*) and three point-mutants of K134 on the chromosomal *nrtR* (namely K134A, K134R, and K134Q, in *Supplementary file 1*). A similar approach was applied to delete the four acetylation pathway genes in *M. smegmatis*, namely *pat* (MSMEG_5458), *cobB* (MSMEG_5175), *ackA* (MSMEG_0784), and *pta* (MSMEG_0783). The resultant mutants denote three single mutants (Δ*pat*, Δ*cobB*, and Δ*ackA*) and a double mutant (Δ*ackA*+Δ*pta*) (*Supplementary file 1*).

### Western blot

Polyclonal anti-serum against MsNrtR was generated by immunizing a rabbit with purified MsNrtR as shown recently (*Gao et al., 2017*). The specificity and sensitivity of the acquired polyclonal antibody was evaluated by western blot and ELISA with pre-immune sera used as a negative control. Subsequently, western blot was conducted routinely (*Gao et al., 2017*). To probe the possible acetylation of the MsNrtR, the anti-acetyl-lysine antibody (Abcam, ab61257) acted as primary antibody. To normalize the relative level of NrtR acetylation, the NrtR concentrations were measured with western blot in which an anti-MsNrtR polyclonal serum was introduced as a primary antibody, as described recently but with minor alteration (*Ren et al., 2016*).

### Non-enzymatic acetylation of NrtR *in vitro*

The MsNrtR protein (10 µg) was incubated with different concentrations of acetyl phosphate (AcP) in Tris-HCl buffer (50 mM [pH 8.0] containing NaCl (150 mM) at 37°C for 2 hr; *Weinert et al., 2013*; *Wang et al., 2017*). Following the separation of the reaction mixture by SDS-PAGE (12%), the acetylated form of MsNrtR protein was detected using western blot with an anti-acetyl-lysine antibody (Abcam, ab61257).

### Determination of intracellular NAD$^+$ and NADH concentrations

Prior to the quantification of the bacterial NAD$^+$/NADH pool, cell counts per optical density at wavelength 600 (OD600) were determined via bacterial plating. In brief, the log-phase cultures of *M. smegmatis* (wild-type and its mutants) were adjusted to OD600 of 1.0, and serially diluted in 10$^5$-folds with fresh 7H9 broth. The resultant bacterial suspension (100 µl) was plated on LB medium,

and kept for around three days at 37°C. Finally, colony counting (each at OD600) was determined to be about $1.0 \times 10^8$ CFU/ml (*Figure 7—figure supplement 1*). The intracellular level of both NAD$^+$ and NADH was determined using a NAD$^+$/NADH kit (Sino Best Biological Technology). Wild-type and mutant strains were collected when their OD$_{600}$ reached around 1.0 (about $1.0 \times 10^8$ CFU/ml, in *Figure 7—figure supplement 1*). Aliquots of bacterial cultures (2–4 ml) were harvested by centrifugation at 4°C for 10 min at 4000 rpm. After discarding the supernatant, the bacterial pellets were washed twice with 1 ml fresh 7H9 medium. Then NAD$^+$ and NADH concentrations in bacterial pellets were extracted and calculated as recommended by the manufacturer.

## RNA isolation, RT-PCR and real-time quantitative RT-PCR

Mid-log phase cultures of *M. smegmatis* and its mutants grown in LB media or 7H9 media were collected for total bacterial RNA preparations. TRIzol reagent (Life Technologies) was used to isolate total RNA. The RNA quality was detected to avoid trace genomic DNA contamination (*Feng and Cronan, 2010*; *Feng and Cronan, 2009b*). First-strand cDNAs were synthesized using a PrimeScript RT reagent Kit with gDNA Eraser (Takara). The final cDNAs were diluted and served as the template for PCR amplification of the *nad* operon-related DNA fragments using specific primers (*Supplementary file 2*). Real-time PCR analysis was performed using SYBR Green Master Mix Reagent (Takara). The 16S rDNA gene served as internal reference and the relative expression levels were calculated using the $2^{-\Delta\Delta CT}$ method (*Livak and Schmittgen, 2001*).

## Enzymatic assays

The Nudix hydrolase activities of NrtR and NrtR$^{Q54E\&K58E\&D60E}$ were purified to homogeneity. The reaction mixture (150 μl) contained 50 mM HEPES (pH 8.2), together with either 5 mM MgCl$_2$ or 0.2 mM ADP-ribose and an appropriate amount of NrtR_ms or NrtR$^{Q54E\&K58E\&D60E}$. After 30 min incubation at 37°C, the reaction was stopped by adding 75 μl of cold 1.2 M HClO$_4$. Reaction products were assayed using HPLC with a column (DiKMA C18-T, 4.6 × 250 mm, 5 μm particle size) at 16°C. The elution conditions were the same as those used before (*Wang et al., 2019*).

## Phylogenetic analysis

NrtR proteins were collated from *Vibrio cholerae*, *Mycobacterium smegmatis* and *Streptococcus suis*. BLASTp (*Johnson et al., 2008*) was used to identify homologs with identity 30% and coverage 30% as cut-offs. 750 homologs of NrtR were found and manually curated in the 9434 RefSeq-archived genomes as of April, 2018, including bacterial and archaeal sequences. To verify the Nudix domain in these homologs, the Protein Families database (Pfam, available at pfam.xfam.org) and the Clusters of Orthologous Groups (COG, available at ncbi.nlm.nih.gov/COG) database were used to identify conserved functional domains (*Tatusov et al., 2000*). Next, protein homologs were further filtered by examining their conserved domain. The proteins carrying only the Nudix domain were removed, and 260 sequences coding for at least two protein domains were kept (Unrooted phylogeny). Subsequently, 38 sequences were further analyzed on the basis of 70% identity at amino-acid sequence level by manual collation (Hierarchical tree). Multiple sequence alignments of protein sequences were produced by the Clustal W program (*Chenna et al., 2003*). The MEGA software was used for construction of a maximum likelihood phylogenetic tree for the NrtR protein family, including bootstrapping with 1000 replicates and drawing of a consensus tree. The number of the corresponding phylogenetic clade reflects the putative evolutionary distance for each node.

## Bioinformatics

The protein sequences of MsNrtR and its homologs from different species were aligned by Clustal Omega (http://www.ebi.ac.uk/Tools/msa/clustalo/), and the final output of the multiple sequence alignments was given processed by the program ESPript 2.2 http://espript.ibcp.fr/ESPript/cgi-bin/ESPript.cgi). Structural modeling for the MsNrtR-DNA was processed using Swiss-Model with SoNrtR-DNA as a structural template (PDB: 3GZ6). The resultant result was given in ribbon structure via PyMol (https://pymol.org/2).

## Acknowledgements

This work was supported by the National Natural Science Foundation of China (31830001, 81772142 and 31570027, YF) and the National Key R & D Program of China (2017YFD0500202, YF). Dr Feng is a recipient of the national 'Young 1000 Talents' Award of China.

## Additional information

### Funding

| Funder | Grant reference number | Author |
| --- | --- | --- |
| National Natural Science Foundation of China | 31830001 | Youjun Feng |
| National Natural Science Foundation of China | 81772142 | Youjun Feng |
| National Natural Science Foundation of China | 31570027 | Youjun Feng |
| Ministry of Science and Technology of the People's Republic of China | 2017YFD0500202 | Youjun Feng |
| Thousand Talents Plan | | Youjun Feng |

The funders had no role in study design, data collection and interpretation, or the decision to submit the work for publication.

### Author contributions

Rongsui Gao, Data curation, Formal analysis, Investigation, Methodology, Writing—original draft; Wenhui Wei, Data curation, Software, Formal analysis; Bachar H Hassan, Jiaoyu Deng, Resources, Software, Formal analysis, Visualization, Methodology, Writing—review and editing; Jun Li, Data curation, Software, Formal analysis, Visualization, Methodology; Youjun Feng, Conceptualization, Resources, Data curation, Software, Formal analysis, Supervision, Funding acquisition, Validation, Investigation, Visualization, Methodology, Writing—original draft, Project administration, Writing—review and editing

### Author ORCIDs

Youjun Feng https://orcid.org/0000-0001-8083-0175

### Decision letter and Author response

Decision letter https://doi.org/10.7554/eLife.51603.025
Author response https://doi.org/10.7554/eLife.51603.026

## Additional files

### Supplementary files

• Supplementary file 1. Strains and plasmids used in this study.
DOI: https://doi.org/10.7554/eLife.51603.020

• Supplementary file 2. Primers used in this study.
DOI: https://doi.org/10.7554/eLife.51603.021

• Supplementary file 3. Nudix family protein in *M. smegmatis* MC$^2$ 155 (*Magni et al., 2004*).
DOI: https://doi.org/10.7554/eLife.51603.022

• Transparent reporting form DOI: https://doi.org/10.7554/eLife.51603.023

### Data availability

All data generated or analysed during this study are included in the manuscript and supporting files.

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
