## [Decision Letter]

Thank you for submitting your work entitled "A Single Regulator NrtR Maintains Bacterial NAD^+^ Homeostasis via Its Acetylation" for consideration by *eLife*. Your article has been reviewed by three peer reviewers, one of whom is a member of our Board of Reviewing Editors, and the evaluation has been overseen by a Senior Editor.

Our decision has been reached after consultation between the reviewers. Based on these discussions and the individual reviews below, we regret to inform you that your work will not be considered further for publication in *eLife*.

The reviewers agreed that the study of NAD biosynthesis and its regulation represents and important, broad reaching area of bacterial metabolism. In general, the experiments are carefully conducted and presented well. However, The role of acetylation of NrtR in regulation of NAD biosynthesis remains unclear *in vivo*. Also there is insufficient evidence presented that the acetyl donor for this process is acetyl-phosphate *in vivo*. These, and other shortfalls mentioned by the reviewers, weaken the manuscript and cast doubt on the conclusions

Reviewer #1:

In most prokaryotes, NAD^+^ is produced either by de novo biosynthesis from tryptophan or through salvage of NAD metabolites. These biosynthetic processes are regulated by three distinct mechanisms, one of which include the activity of a family of Nudix-related transcriptional regulators (NrtRs). The authors set out to characterise this mode of regulation in mycobacteria.

Key findings:

1) Using computational approaches, the authors identify an NrtR binding palindromic site proximal to the locus encoding the first three enzymes in the NAD^+^ biosynthesis pathway. This site is located between the *nrtR* gene and the nadA/B/C operon.

2) They describe the phylogenetic distribution of NrtR-like homologues and highlight some differences in domain organisation and possible functional consequences. This is followed by homology modelling of NrtR bound to DNA and description of the resiu.

3) They purify the Mycobacterium smegmatis NrtR (confirmed by SDS-PAGE, chemical cross-linking and MS) and demonstrate that it binds in a concentration-dependent manner to the putative regulatory sequence. They further construct six point mutations and demonstrate that these residues are individually essential for NrtR DNA binding activity.

4) They demonstrate that ADP-ribose, a toxic breakdown product of NAD^+^, inhibits NrtR DNA binding activity, although, NrtR does not have an ADP-ribose pyrophosphohydrolase activity.

5) A mutant lacking NrtR displayed increased expression of *nadA* and elevated levels of NAD^+^/NADH – confirming NrtR as a negative repressor of NAD biosynthesis, regardless of growth phase. NrtR also autoregulates its own biosynthesis.

6) The authors demonstrate that NrtR is acetylated, predominantly at the K134 residue, mutagenesis of this residue to remove the acetylation effect resulted in variances in DNA binding – not sure what this means, not quantified by SPR, as with their earlier experiment.

7) Acetylation by a non-enzymatic acetyl phosphate-dependent mechanism seems to be responsible for acetylation.

8) in vivo, acetylation was required for NrtR-dependent regulation of the *nadABC* operon and NAD+/NADH biosynthesis.

Generally the study is conducted well, with a careful experimental approach. The following should be addressed.

1) Much of the conclusions described in the bioinformatics sections about domains and putative function appear to be speculation as there are no references to experimental validation. This section should be shortened.

2) All electrophoretic gel mobility assays are missing cold competitive controls. This is necessary to ensure specificity and should be done.

3) For Figure 5—figure supplement 1, some graphic representation of the westerns is required to make comparisons. Density scans of the bands should be carried out over multiple experiments.

4) Whilst not essential to the entire story, the dimerization of acetylation defective mutants should have been assessed. Perhaps acetylation affects protein-protein interactions.

Reviewer #2:

The regulation of NAD biosynthesis in bacteria occurs through the NrtR and/or NadR regulators. The NrtR transcriptional regulator regulates the expression of the NAD biosynthetic genes in organisms such as Corynebacterium. In mycobacteria there are both NadR and NrtR homologs, depending on the species and the regulation of mycobacterial NAD biosynthesis by NrtR has been discussed based on work done in other bacteria (see Bi et al., 2010. In this work the authors confirm that NrtR regulates expression of the *nad* genes in *M. smegmatis* and that, similar to the situation in some NrtR homologs, ADP-ribose binds to the protein, affecting DNA binding, but is not hydrolyzed by it. Deletion of the *nrtR* homolog in *M. smegmatis* modestly affects NAD(H) and *nadA* transcript levels. The really novel aspect of this work is the finding that recombinantly expressed protein is acetylated. The acetyl donor is speculated to be acetyl-phosphate but there is little evidence to support this except for the fact that this reactive metabolite, at high concentrations, can acetylate to protein*in vitro*. The role of acetylation is speculated to be regulation of protein binding but it should be noted that there is no evidence provided in this work that protein acetylation is ever regulated. Recombinantly expressed protein is always found to be acetylated whereas mutants of the reactive lysine, although they seem to bind cognate DNA in EMSA experiments (contrary to what the authors state), seem to be inactive in cells returning levels of transcript and NAD(H) to that found in knockout mutant cells.

Overall, the experiments are elegantly performed with well-done biochemistry. It certainly is interesting that the protein is always acetylated but the relevance of this acetylation to regulation of NAD biosynthesis is not known.

Reviewer #3:

Goal: demonstrate that the Nudix-related transcriptional regulator NrtR (MSMEG_3198) controls NAD homeostasis in *M. smegmatis* via acetylation of a lysine group at position 134.

The authors found a 23 bp palindromic sequence between nrtR and *nadABC*, assert that it binds to NrtR yet do not shown evidence to support this claim.

The authors constructed a phylogenetic tree of 260 Nudix protein family representatives including only 2 Nudix proteins from M. smegmatis. It would have been useful to include the Nudix proteins from other mycobacterial species such as *M. tuberculosis, M. avium, M. leprae*, or *M. marinum*.

The authors tested whether NrtR had the predicted ADP-ribose pyrophosphohydrolase activity but could not detect any.

The authors constructed a nrtR KO in *M. smegmatis* and showed by RT-PCR that deletion of nrtR increased *nadA* expression by 2-6 fold. The authors also measured NAD+ and NADH in wt and nrtR KO strains and observed less than 2-fold increase in the cofactor levels. This experimental design lacks the requisite precision to draw a meaningful conclusion as they reported levels of NAD+ and NADH as pmol/106 CFU but they did not plate for CFUs in their experiment. They only estimated CFUs based on OD while admitting that there could be a 2-fold variation in their CFU estimation. Therefore, the results cannot support the conclusion that NrtR regulates NAD+ and NADH levels. Additionally, nrtR deletion does not modify the NADH/NAD+ ratio as both cofactors are altered at the same rate further indicating that NrtR has no role in maintaining the redox status of mycobacterial cells.

The authors demonstrated that *M. smegmatis* NrtR K134 lysine residue is acetylated. They constructed *M. smegmatis* mutants where the lysine residue was replaced by an alanine, a glutamine or an arginine group. These mutants had similar phenotypes as the nrtR KO strain. The authors concluded that "acetylation of K134 is a prerequisite for NrtR to regulate homeostasis of NAD+ in Mycobacterium". This conclusion is not substantiated by the data provided.

The authors should have compared the data obtained from this study with NadR, a known regulator of NAD biosynthesis. I would not recommend this for publication.

[Editors’ note: what now follows is the decision letter after the authors submitted for further consideration.]

Congratulations, we are pleased to inform you that your article, "A single regulator NrtR controls bacterial NAD^+^ homeostasis via its acetylation", has been accepted for publication in *eLife*.

Prokaryotic NAD+ biosynthesis is regulated by three distinct mechanisms, one of which includes the activity of a family of Nudix-related transcriptional regulators (NrtRs). Your work describes the role of NrtR in modulating the activity of the biosynthetic operon for NAD+ biosynthesis in mycobacteria. The finding that NrtR is acetylated by acetyl-phosphate and that this acetylation status affects NAD+ homeostasis is novel and has important implications for studies on NAD+, and related cofactor, biosynthesis in other prokaryotes. Metabolic enzymes/pathways in bacteria are fast gaining traction as novel drug targets, and how these pathways are regulated is central to the discovery of new antimicrobials. Hence, your work also has important implications for bacterial, target-driven drug discovery approaches. Further dissection of the interplay between this regulatory mechanism and other modalities of NAD biosynthesis should form an important component of future work in this area of bacterial metabolism.

*Reviewer #1:*

The revised manuscript by Gao and colleagues attempts to improve upon the earlier submission and address concerns that were raised during review. This manuscript reports a study that was aimed to investigate the regulation of NAD^+^ biosynthesis in bacteria, with a focus on the Nudix-related transcriptional regulators (NrtRs). In mycobacteria, the authors reported the presence of palindromic binding sites for a regulator upstream of the *nadA/B/C*operon and demonstrated that the regulator, NrtR, bound this in a somewhat sequence specific manner that is affected by ADP-ribose, a toxic breakdown product of NAD^+^. The authors also demonstrated that NrtR is acetylated by a non-enzymatic acetyl phosphate-dependent mechanism. The specific concerns that were raised and how these have been addressed are outlined below.

1) Long and complicated exposition of bioinformatics: This has been shortened and is more succinct. In general, some of the writing still needs attention. Tenses are mixed up. In the Results section and in the Abstract, results need to be reported in the past tense. The phyologenetics has also been appropriately revised.

2) EMSA controls: They attempted to address this through adding non-specific controls. This is acceptable but needs to be displayed with the specific positive controls on the same gel. Figure 3—figure supplement 2 needs to be amended to add a lane of specific DNA, or it should be removed. Binding of NrtA to its own promoter is now convincingly shown in Figure 3—figure supplement 3C.

3) Quantification of westerns: This is addressed in an acceptable manner, please add statistics to the figure.

4) Dimerization status of mutant proteins: Addressed.

5) High concentrations of ADP-ribose: The authors make a compelling argument, based on evidence from the literature.

6) Acetylation (mechanism and role *in vivo*): This was a substantive comment from all reviewers. The authors have attempted to address. They mutate the enzymatic mechanism of acetylation through deletion of the AcP pathway in Mycobacterium smegmatis and confirm acetylation status. The evidence is not definitive but does bring greater clarity. The conclusions regards acetylation could be stated more carefully, in cognisance of the knowledge gaps that still exist.

Language has improved substantively but it can still be polished further. The last sentence of the Abstract is not clear. What does "it" refer to? Most likely the mechanism of regulation but a more careful statement of the overall conclusions is required. Another round of careful reading and editing should address these minor issues.

*Reviewer #2:*

The authors have justified their conclusion that levels of acetylation of NtrR by acetyl-phosphate affect NAD homeostasis in Mycobacterium. Thus, the finding that NtrR is acetylated and that the level of acetylation modestly affects transcription of the NAD biosynthetic genes and NAD levels is novel and an important finding.

---

## [Author Response]

In this revision, we have invited an expert in *Mycobacterium*acetylation, Prof. Jiaoyu Deng to help improve this manuscript. More importantly, we have addressed all the questions appropriately, modified the manuscript extensively, and added all the new data (*in vitro* and i*n vivo*) requested by three referees.

In brief, all the new data are included as follows: 1) phylogenetic tree has been replaced, including NrtR homologs of different Mycobacterium species; 2) the negative control of EMSA has been provided; 3) we added the relative quantification results of NrtR mutants in the altered DNA-binding ability; 4) we provided our platting data used to measure CFU, which is helpful in declaring the misunderstanding of referee 3 on this issue; 5) we expanded the generality of NrtR acetylation by examining NrtR from two more microbes, Gram-negative *Vibrio cholerae* and Gram-positive *Streptococcus suis*); 6) we constructed mutants of M. smegmatis whose AcP route is inactivated, namely a single mutant of *ΔackA* and a double mutant of *ΔackA Δpta*. As Weinert et al stated (Weinert et al., 2013), we found that the NrtR acetylation is significantly impaired in the double mutant on the certain growth condition (such as the addition of exogenous acetate or glucose, Figs 6H and I). It constitutes *in vivo* evidence that NrtR is acetylated via Acp-dependence (Figures 6H and I), rather than the enzymatic action of Pat/CobB (Figures 6B-G).

Reviewer #1:

In most prokaryotes, NAD+ is produced either by de novo biosynthesis from tryptophan or through salvage of NAD metabolites. These biosynthetic processes are regulated by three distinct mechanisms, one of which include the activity of a family of Nudix-related transcriptional regulators (NrtRs). The authors set out to characterise this mode of regulation in mycobacteria.Key findings:1) Using computational approaches, the authors identify an NrtR binding palindromic site proximal to the locus encoding the first three enzymes in the NAD+ biosynthesis pathway. This site is located between the nrtR gene and the nadA/B/C operon.2) They describe the phylogenetic distribution of NrtR-like homologues and highlight some differences in domain organisation and possible functional consequences. This is followed by homology modelling of NrtR bound to DNA and description of the resiu.3) They purify the Mycobacterium smegmatis NrtR (confirmed by SDS-PAGE, chemical cross-linking and MS) and demonstrate that it binds in a concentration-dependent manner to the putative regulatory sequence. They further construct six point mutations and demonstrate that these residues are individually essential for NrtR DNA binding activity.4) They demonstrate that ADP-ribose, a toxic breakdown product of NAD+, inhibits NrtR DNA binding activity, although, NrtR does not have an ADP-ribose pyrophosphohydrolase activity.5) A mutant lacking NrtR displayed increased expression of nadA and elevated levels of NAD+/NADH – confirming NrtR as a negative repressor of NAD biosynthesis, regardless of growth phase. NrtR also autoregulates its own biosynthesis.6) The authors demonstrate that NrtR is acetylated, predominantly at the K134 residue, mutagenesis of this residue to remove the acetylation effect resulted in variances in DNA binding – not sure what this means, not quantified by SPR, as with their earlier experiment.7) Acetylation by a non-enzymatic acetyl phosphate-dependent mechanism seems to be responsible for acetylation.8) In vivo, acetylation was required for NrtR-dependent regulation of the nadABC operon and NAD+/NADH biosynthesis.

We are thankful to reviewer 1 for detailed summary of key findings in this study. As for the 6^th^ concern raised by reviewer 1, it is because that i) two different approaches (SPR and EMSA) are found to be effective in assaying the NrtR-DNA interaction; ii) EMSA is much cheaper and convenient method when compared to that of SPR.

Generally the study is conducted well, with a careful experimental approach. The following should be addressed.1) Much of the conclusions described in the bioinformatics sections about domains and putative function appear to be speculation as there are no references to experimental validation. This section should be shortened

We have rephrased this section appropriately. Also, references are cited accordingly.

2) All electrophoretic gel mobility assays are missing cold competitive controls. This is necessary to ensure specificity and should be done.

It is a good suggestion. Technically, cold probe is only suitable in the EMSA with isotope-labeled DNA probe. However, our assay is based on EB staining of PAGE. That is why cold probe can’t work here. Equivalently, we introduced an unrelated DNA probe, *vprA* promoter as a negative control, which effectively verify the specificity of NrtR-cognate interaction (new Figure 3—figure supplement 2).

3) For Figure 5—figure supplement 1, some graphic representation of the westerns is required to make comparisons. Density scans of the bands should be carried out over multiple experiments.

As reviewer 1 suggested, no less than 3 independent trials of western blot have been calculated. Here, relative quantification results of NrtR protein in the DNA-binding affinities were plotted (new Figure 5—figure supplement 1G).

4) Whilst not essential to the entire story, the dimerization of acetylation defective mutants should have been assessed. Perhaps acetylation affects protein-protein interactions.

It is a good point, because that acetylation can affect protein-protein interaction in certain cases. However, our results of gel filtration experiments revealed that all the K134 mutant NrtR proteins (K134A, K134Q, and K134R) still possess the ability to form dimer in solution, ruling out this possibility (new Figure 5—figure supplement 1B).

Reviewer #2:

[…] Overall, the experiments are elegantly performed with well-done biochemistry. It certainly is interesting that the protein is always acetylated but the relevance of this acetylation to regulation of NAD biosynthesis is not known.

We do appreciate reviewer 2 for the positive overall judgement on this work. In the revision, we have added new data, that is solid in vivo evidence that such kind of acetylation is dependent on the non-enzymatic AcP action, rather than the enzymatic form of Pat/CobB. It is unusual, but not without precedent, because that AcP-dependent acetylation seems common in *E. coli* as Weinert and coworkers described (Weinert et al., 2013). In fact, protein acetylation has been well known as a prevalent form of post-translational modifications and plays multiple roles in bacterial physiology, augmenting cross-talks amongst different metabolisms at multiple levels. NAD^+^ refers to an essential vitamin involving in central metabolic activities. Evidently, the deacetylase CobB also requires NAD^+^ for its enzymatic activity (Sang et al., 2016) and links to the signaling of the second messenger c-di-GMP (Xu et al., 2019). To the best (but not limited to) of our knowledge, NAD^+^ has already been connected with protein acetylation in mammals. Not only does cytosolic NAD^+^ level involve protein acetylation (Marcu et al., 2014), but its depletion of NAD^+^ connects with p65 acetylation via the elevation of nuclear factor-kappaB transcription (Kauppinen et al., 2013). In *Saccharomyces cerevisiae*, it has been elucidated that a functional link occurs between NAD^+^ homeostasis NatB-mediated protein acetylation (Croft et al., 2018). Together with the aforementioned literature, we can conclude that i) the observation of NAD^+^ homeostasis in connection with NrtR acetylation is physiologically reasonable; 2) it represents a first example that acetylation of NrtR controls NAD^+^ metabolism at least in the prokaryotic domain of life. Finally, we hope reviewer 2 can understand it and respect the novelty on this standing.

Reviewer #3:

Goal: demonstrate that the Nudix-related transcriptional regulator NrtR (MSMEG_3198) controls NAD homeostasis in M. smegmatis via acetylation of a lysine group at position 134.The authors found a 23 bp palindromic sequence between *nrtR* and *nadABC*, assert that it binds to NrtR yet do not shown evidence to support this claim.

Technically, a target palindrome sequence is too short to directly be applied in the *in vitro* EMSA experiment. In general, a DNA probe containing this palindrome is applied in EMSA assays. As described by different groups (Huang et al., 2009; Rodionov et al., 2008; Wang et al., 2019) with little change, we used a short probe 57bp covering the 23 palindrome in our SPR (Figure 3D) and EMSA assays (Figure 3C and new Figures 3—figure supplement 3C and Figure 5—figure supplement 1C). Two different approaches convincingly confirm this DNA-NrtR interaction. This binding is specific in that NrtR can’t bind to its unrelated *vprA* promoter (new Figure 3—figure supplement 2). More importantly, this physical binding of NrtR and cognate DNA has physiological role in the regulated expression *nad* operon and level of cytosolic NAD^+^ (NADH) pool (Figure 4). Therefore, we believe our result is solid and convincing, which is fully agreement with those reported by others (Huang et al., 2009; Rodionov et al., 2008; Wang et al., 2019).

The authors constructed a phylogenetic tree of 260 Nudix protein family representatives including only 2 Nudix proteins from M. smegmatis. It would have been useful to include the Nudix proteins from other mycobacterial species such as M. tuberculosis, M. avium, M. leprae, or M. marinum.

In fact, four NrtR-like proteins, Rv1593c [*Mycobacterium tuberculosis* H37Rv], MAP4_2560 [*Mycobacterium avium*subsp. paratuberculosis MAP4], ML1224 [*Mycobacterium leprae* TN], MMAR_2390 [*Mycobacterium marinum* M] have been included in the unrooted tree (Figure 2A). In addition, we found that their sequences were highly similar to MSMEG_3198 [*Mycolicibacterium smegmatis* MC^2^ 155] and MSMEI_3116 *[Mycobacterium smegmatis*str. MC^2^ 155] via BLASTp-based assays. For example, Rv1593c is 79% identical to MSMEG_3198. Based on this scenario, they were compactly grouped into a clade.

Next, we concluded these four protein sequences to remove redundant homologs for the inferred phylogeny shown in old Figure 2B (cut-offs: 70% identity). However, we fully agreed with the reviewer’s critical suggestion, considering the important representativeness of *M. avium, M. leprae, M. marinum* and *M. tuberculosis* for*Mycobacterium* strains.

Here, we followed reviewer 3’s comment to re-introduce them into our revised phylogenetic tree (new Figure 2B).

The authors tested whether NrtR had the predicted ADP-ribose pyrophosphohydrolase activity but could not detect any.

Yes, we did. Unfortunately, we can’t detect enzymatic activity. This is an evolutionary relic (Huang et al., 2009; Rodionov et al., 2008; Wang et al., 2019). In fact, a similar scenario is seen with NrtR of *Streptococcus suis* (Wang et al., 2019).

The authors constructed a *nrtR* KO in *M. smegmatis* and showed by RT-PCR that deletion of *nrtR* increased *nadA* expression by 2-6 fold. The authors also measured NAD+ and NADH in wt and nrtR KO strains and observed less than 2-fold increase in the cofactor levels. This experimental design lacks the requisite precision to draw a meaningful conclusion as they reported levels of NAD+ and NADH as pmol/10^6^ CFU but they did not plate for CFUs in their experiment. They only estimated CFUs based on OD while admitting that there could be a 2-fold variation in their CFU estimation. Therefore, the results cannot support the conclusion that NrtR regulates NAD^+^ and NADH levels.

First, we would like to thank reviewer 3 for the summary of nrtR KO results. It seems likely that misunderstanding or omitting the plating experiment by the reviewer 3 is due to my simplified statement in the Materials and methods section in last version. In this revision, we have added detailed information on how I do the bacterial cell plating (subsection “Determination of intracellular NAD+ and NADH concentrations”). The plating counts appeared in new Figure 7—figure supplement 1. We thereby expect that reviewer 3 can consider it on this standing, and our result is reasonable.

Additionally, *nrtR* deletion does not modify the NADH/NAD^+^ ratio as both cofactors are altered at the same rate further indicating that NrtR has no role in maintaining the redox status of mycobacterial cells.

We have read careful several literatures related to NAD metabolism in *Mycobacterium* from the research group of reviewer 3. We found them informative and cited them appropriately in the section of Introduction. It was described as follows: “Earlier microbial study by Vilcheze et al. (Vilcheze et al., 2005) indicated that the removal of *ndhII*, a type II NADH dehydrogenase-encoding gene, enhances the intracellular NADH/NAD^+^ ratio, giving phenotypic resistance to the front-line anti-TB drug isoniazid (INH) and its related drug ethionamide (ETH). Subsequently, the de novo and salvage pathways of NAD^+^ is proposed to exhibit potential of being anti-TB drug targets (Vilcheze et al., 2010)“.

Presumably, the checkpoint of NrtR is the mixed pool of cytosolic NAD^+^ and NADH, rather than the ratio of NAD^+^ /NADH (Huang et al., 2009; Rodionov et al., 2008; Wang et al., 2019). In contrast, the ratio of NAD^+^ /NADH in *E. coli* is determined by the reversable reduction reaction of NAD. This reaction is catalyzed by PntA/B and UdhA in *E. coli* (Anderlund et al., 1999; Boonstra et al., 1999; Sauer et al., 2004). In *M. smegmatis*, the homologs appear as follows: MSMEG_0110 (75% similarity) for *pntA*, MSMEG_0109 (77.7% similarity) for *pntB*, and MSMEG_2748 (58.1% similarity) for *udhA*. However, the aforementioned three genes are not cognate targets of NrtR at all. Therefore, the available literature and reasonable analysis argues against the statement by reviewer 3. Obviously, the ratio of NADH/NAD^+^ ratio should be the checkpoint of the reaction by the three enzymes of PntA, PntB and UdhA.

The authors demonstrated that *M. smegmatis* NrtR K134 lysine residue is acetylated. They constructed *M. smegmatis* mutants where the lysine residue was replaced by an alanine, a glutamine or an arginine group. These mutants had similar phenotypes as the *nrtR* KO strain. The authors concluded that "acetylation of K134 is a prerequisite for NrtR to regulate homeostasis of NAD^+^ in *Mycobacterium*". This conclusion is not substantiated by the data provided.The authors should have compared the data obtained from this study with NadR, a known regulator of NAD biosynthesis. I would not recommend this for publication.

We do respect the criticism of reviewer 3. However, to the best of our knowledge, our *in vitro* and *in vivo* data have reached the high standing level in the acetylation filed. Moreover, we added new in vivo data (i.e., removal of AcP pathway-encoding genes Pta/AckA), supporting that NrtR proceeds in AcP-dependent acetylation (new-Figure 6). This is completely consistent with our former conclusion. As for the suggestion of “comparison with NadR, a known regulator”, we did it. However, it is not practically feasible, because the only available two literatures of NadR from *E. coli* and *Salmonella* is an *in vitro* EMSA description without any physiological data of NAD^+^ level (Penfound and Foster, 1999; Raffaelli et al., 1999).